# Brain-wide microstrokes affect the stability of memory circuits in the hippocampus

Hendrik Heiser [1,2], Filippo Kiessler [3], Adrian Roggenbach [1,2], Victor Ibanez[1,2], Martin Wieckhorst[1,2], Fritjof Helmchen [1,2,4], Julijana Gjorgjieva [3] & Anna-Sophia Wahl [1,2,5,6] ✉

Cognitive deficits affect over 70% of stroke survivors, yet the mechanisms by which multiple small ischemic events contribute to cognitive decline remain poorly understood. In this study, we employed chronic two-photon calcium imaging to longitudinally track the fate of individual neurons in the hippocampus of mice navigating a virtual reality environment, both before and after inducing brain-wide microstrokes. Our findings reveal that, under normal conditions, hippocampal neurons exhibit varying degrees of stability in their spatial memory coding. However, microstrokes disrupted this functional network architecture, leading to cognitive impairments. Notably, the preservation of stable coding place cells, along with the stability, precision, and persistence of the hippocampal network, was strongly predictive of cognitive outcomes. Mice with more synchronously active place cells near important locations demonstrated recovery from cognitive impairment. This study uncovers critical cellular responses and network alterations following brain injury, providing a foundation for novel therapeutic strategies preventing cognitive decline.

Although many stroke survivors develop forms of cognitive decline[1,2], the pathomechanisms how cognitive decline emerges, even if cognitive brain areas are not directly affected, are not understood, nor are there any specific treatment options available. In particular, the accumulation of multiple, smaller ischemic events−with other obvious symptoms of stroke, such as motor impairment lacking−does not lead to acute cognitive impairment, but cognitive decline develops in months and years after the ischemic events[3]. The hippocampus, a relay station for cognition and memory processing, is particularly susceptible to ischemic events[4−6], with neuronal death occurring already after a brief episode of ischemia[7,8]. Although this vulnerability of the hippocampus has been discussed in the context of its high plasticity, the anatomical connection to many brain areas and its vascularization[8,9], it is not understood how individual neurons and functional networks in

the hippocampus react to brain-wide injuries and how they rewire and recode to maintain their function. Several types of neurons coding for distinct memory functions have been identified: O'Keefe and Dostrovsky[10] discovered 'place cells' (PCs) in 1971 which respond specifically to the current location of the animal, but have been also discussed[11] to contain compressed representation of contextual, sensory and episodic experiences during exploration of an environment. While the importance of 'place cells'[10] for memory formation and maintenance has been extensively demonstrated, it remains elusive, how place cells react individually or in ensembles to the induction of multiple microstrokes distributed throughout the brain. Revealing the functional impact of microstrokes on the single-cell level and on network tuning properties is of high translational value to better link neuropathological features after ischemic events to cognitive decline

[1]Brain Research Institute, University of Zurich, Winterthurerstrasse 190, 8057 Zurich, Switzerland. [2]Neuroscience Center Zurich (ZNZ), University of Zurich, Zurich, Switzerland. [3]School of Life Sciences, Technical University of Munich, Maximus-von-Imhof-Forum 3, 85354 Freising, Germany. [4]University Research Priority Program (URPP), Adaptive Brain Circuits in Development and Learning, University of Zurich, Zurich, Switzerland. [5]Institute for Stroke and Dementia Research (ISD), LMU University Hospital, LMU Munich, Feodor-Lynen-Strasse 17, 81377 Munich, Germany. [6]Department of Neuroanatomy, Institute of Anatomy, Ludwigs-Maximilians-University, Pettikoferstrasse 11, 80336 Munich, Germany. ✉e-mail: AnnaSophia.Wahl@med.uni-muenchen.de

and to identify new targets for treatment strategies preventing cognitive deficits.

Here, we present an innovative approach, where we studied the effect of brain-wide microstrokes on individual neurons and functional networks in CA1 of the hippocampus in mice performing a cognitive task. Using chronic two-photon calcium imaging in the hippocampus in mice navigating in a virtual reality corridor we could follow the fate of individual neurons in the healthy condition but also several weeks after the induction of disseminated cerebral microstrokes. We reveal that brain-wide microstrokes disrupt individual neuronal coding, functional hippocampal network architecture and induce cognitive decline. Our findings highlight the importance of understanding fundamental reorganization principles in the hippocampus on a cellular resolution level in relation to measurable cognitive deficit parameters: Analyzing the cellular and network responses after stroke, we could show that the cognitive outcome of animals is critically related to the individual stability of place cells or sub-networks of active surviving neurons with similar spatial tuning to maintain memory function and prevent cognitive decline.

## Results

### Microstrokes impair spatial memory and place cell stability

To identify the function of individual neurons during a spatial navigation task, we performed chronic two-photon calcium imaging of the same hippocampal network in the healthy condition (Fig. 1A) and after stroke (Fig. 1B). Mice expressing the calcium indicator GCaMP6f in CA1 were trained to navigate head-fixed in a virtual reality (VR) corridor (Fig. 1A), while we simultaneously recorded calcium signals of neuronal populations in CA1 through a chronically implanted glass window 5 days before and up to 28 days after stroke surgery (Fig. 1C, D).

We induced microstrokes by injecting fluorescent microspheres (20 μm diameter) unilaterally in the internal carotid artery, inducing disseminated microstrokes (Fig. 1B, Figure S1A, B). Microspheres were found directly visible under the hippocampal window (Figure S1A) in 4 out of 20 animals. We performed histological analysis to identify the microsphere distribution in the mouse brains (Figure S1B) and to quantify the lesion volume: Lesion volume and microsphere load were strongly correlated (Spearman's $\rho = 0.84$, $p < 0.001$; linear regression model: slope = 0.001, $R^2 = 0.94$; Figure S1E). Most microspheres and lesions were located in the neocortex (spheres: $39.3 \pm 2.2\%$, lesions: $31.3 \pm 4.3\%$), and in the hippocampus (spheres: $13.8 \pm 1.7\%$, lesions: $14.2 \pm 3.5\%$; Figure S1F), but also subcortically, e.g. in the thalamus and striatum (Figure S1F).

We trained mice to correctly identify four reward zones in the VR track where they received a water reward (Fig. 1E). While naïve mice searched for water rewards in the entire corridor, expert mice learned to lick for water primarily in reward zones ( > 60% licks within reward zones, Fig. 1E). Microstrokes disrupted this licking pattern so that animals again randomly licked throughout the track, suggesting that mice lost the learned ability to locate themselves in the corridor. Notably, microstrokes did not affect the licking rate itself, as sham and stroke mice did not show significant differences before and after stroke induction (Figure S2E, F).

The more microspheres we found post-mortem in the brain, the worse was the performance of the animals in the VR corridor, particularly early (within 7 days) after stroke ($r = -0.53$, $p = 0.01$; Pearson correlation, Fig. 1F). Similarly, after induction of microstrokes animals with poor task performance displayed lower mean firing rates of CA1 neurons, a lower place cell ratio (percentage of all imaged cells that were classified as place cells, see methods), and a reduced stability of place cells to remain place cells across trials in individual sessions ("within-session stability", Fig. 1G). Animals with microstrokes displayed only transient, very minor neurological deficits on day 2 after stroke (Figure S2A, B), whereas we observed no impairment of task-

relevant motor abilities (locomotion, Figure S2C, D; or licking, Figure S2E) compared to sham animals. A generalized linear model revealed no significant association between the number of microspheres in a distinct brain region and the cognitive performance in the VR corridor early after stroke (Figure S1H) or metrics of neural coding (Figure S1I), suggesting that the number of accumulated microstrokes and thus the affected brain volume was more critical for the cognitive performance than the location of individual microstrokes. As our spatial navigation task involved in particular visual cues in the VR-setup to identify reward zones, we also analyzed the relative amount of microspheres in visual areas (following the Allen Brain Atlas as reference). We found no significant correlation between the number of microspheres stuck in visual areas and the task performance after stroke, suggesting no direct link between damage to the visual system and task performance (Figure S1G).

Our experimental setup allowed us to track the activity of the same individual neurons before and after microstrokes for several weeks while animals performed the spatial navigation task. We could thus classify neurons according to the stability of their spatial tuning: We identified stable place cells, which were active when the mouse was at the same position in the VR during 5 consecutive days pre-stroke, whereas unstable place cells remapped their spatial field (Fig. 1H, I). We could also identify non-coding cells, which did not meet the place cell criteria[12] within imaging sessions (Fig. 1H, I). When we compared the fractions of these three functional classes across experimental phases (healthy pre-stroke, early and late after stroke) and between stroke and sham animals (Fig. 1J-L), we found that animals with microstrokes had significantly fewer stable place cells, both early and late after stroke compared to sham animals (fraction of stable place cells early after stroke: Sham: $14.7 \pm 3.3\%$ versus Stroke: $2.9 \pm 1.3\%$, $p = 0.005$; fraction of stable place cells late after stroke: Sham: $24.4 \pm 4.5\%$ versus Stroke: $10.8 \pm 3.2\%$, $p = 0.025$; Fig. 1J). Stroke mice also had significantly more non-coding cells than sham animals (fraction of non-coding cells late after stroke: Sham: $64.1 \pm 5.5\%$ versus Stroke: $79.3 \pm 4.4\%$, $p = 0.046$; Fig. 1L), while the fraction of unstable place cells remained unchanged for both groups (Fig. 1K), indicating that microstrokes affected the functional properties of surviving neurons and the stability of spatial tuning.

### Microstrokes induce place cell turn-over

As we had identified a loss of stable place cells after stroke, we next investigated whether individual surviving neurons had maintained or switched their functional class for spatial tuning over time from the healthy to the disease state (Fig. 2A). As functional remapping of hippocampal networks is known to be naturally highly dynamic[12–14], we first quantified the probabilities of place cells and non-coding cells to be a place cell on the next day in healthy networks, and compared these to chance level, simulated by a distribution where cell classes were randomly shuffled, such that a transition occurred more often than chance given a positive $\Delta P$ (with $\triangle P = P_{true} - P_{shuffle}$), and less often than chance with a negative $\Delta P$. We found that place cells were more likely to remain place cells than chance level, and noncoding cells less likely to become place cells than the chance level of 0 (place cell to place cell (PC-PC) transitions: $13.5 \pm 2.4\%$, $p < 0.001$; noncoding cell to place cell transitions (NC-PC): $-1.3 \pm 0.4\%$, $p = 0.004$; Fig. 2B). Once the animal learned the task, the corresponding neuronal network consolidated. Neurons were attributed to distinct functions within the network, making it less likely that these cells switch their dedicated function. We found that microstrokes transiently induced turn-over[15] of neurons with dedicated function: While there was a higher probability in sham animals of place cells to stay place cells and a low probability of non-coding cells to become place cells over time (Fig. 2C), in stroke mice the probability of place cells keeping and non-coding cells changing their functional identities were at chance level in the early phase after stroke (Stroke group early post-stroke: PC-PC

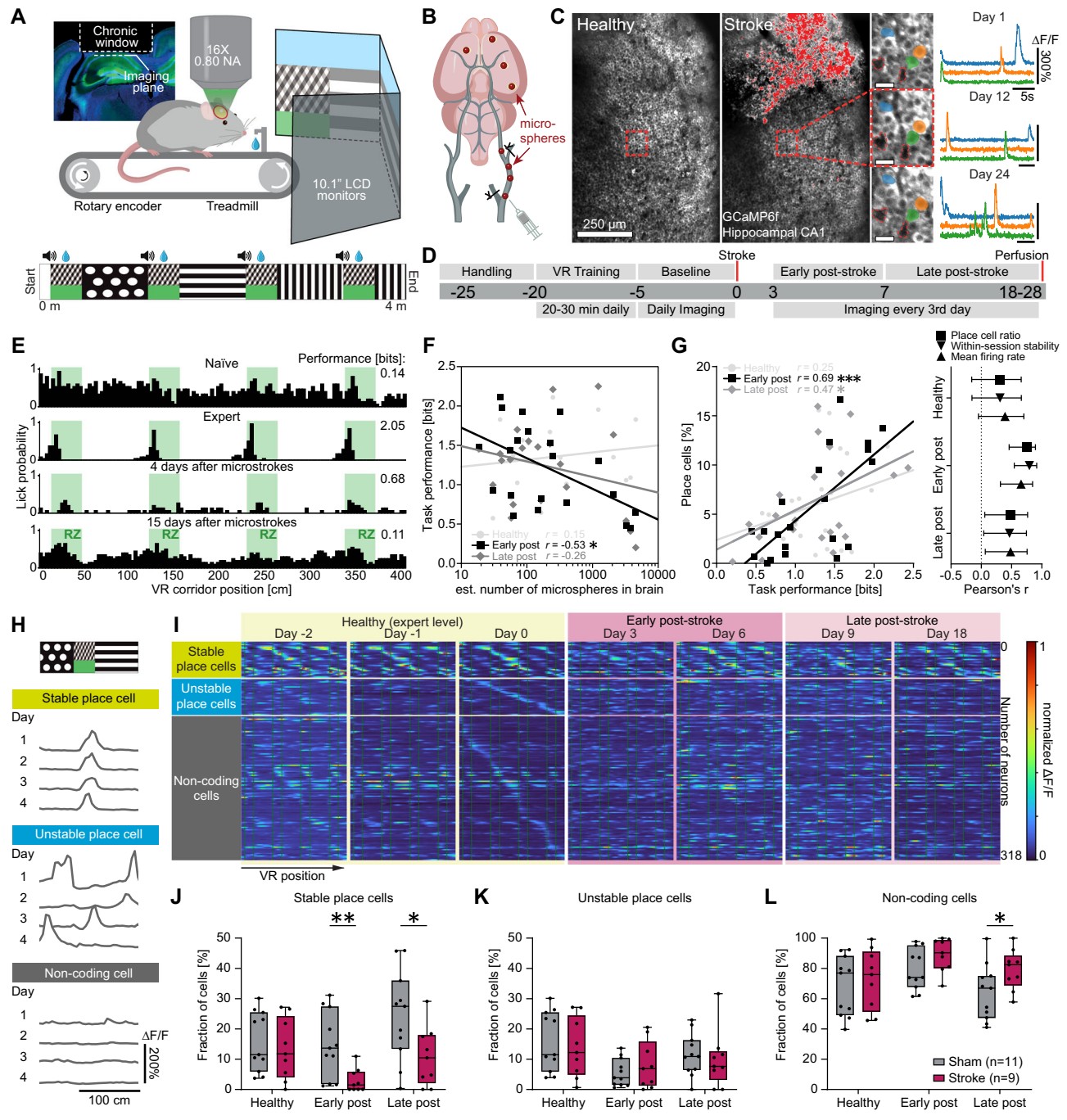

transitions: 1.4 ± 2.4%, $p = 0.580$; NC-PC transitions: -0.9 ± 0.7%, $p = 0.219$; Fig. 2C left), and significantly lower than in sham mice (PC-PC transitions: Sham group versus Stroke group: $p < 0.001$; Fig. 2C left), indicating that neurons were randomly assigned to different functions immediately after stroke. Notably, the turn-over was only transient and the network re-consolidated during the late post-stroke phase, where stroke animals had place cells retaining their function above chance level and non-coding cells switching function below chance level (Stroke group late post-stroke: PC-PC transitions: 16.2 ± 3.8%, $p = 0.004$; NC-PC transitions: -0.8 ± 0.3%, $p = 0.025$; Fig. 2C right), similar to sham animals and the healthy condition (Fig. 2B).

We next examined the shift of the three identified functional cell classes (noncoding cells, unstable and stable place cells) when comparing the healthy and post-stroke condition (Fig. 1H, I). Indeed, in

sham animals stable place cells (sPC, yellow) and noncoding cells (NC, grey) preferably maintained their function compared to chance level (Sham group: NC-NC transitions: 6.8 ± 2.1%, $p = 0.009$; sPC-sPC transitions: 14.7 ± 2.2%, $p < 0.001$, Fig. 2D). The transition to other functional classes was less likely (Sham group: NC-sPC transitions: -6.4 ± 2.1%, $p = 0.012$; sPC-NC transitions: -15.0 ± 1.8%, $p < 0.001$). This was in strong contrast to stroke animals, where transition probabilities between all three classes were at chance level early after stroke (Stroke group: e.g. sPC-sPC transitions: 2.6 ± 1.6%, $p = 0.134$; sPC-NC transitions: -2.8 ± 1.7%, $p = 0.142$, Fig. 2D), and stable place cells retained their function significantly less often ($p = 0.004$), while becoming non-coding more often than in sham animals ($p < 0.001$). Interestingly, sham animals showed a tendency of unstable place cells (uPC, blue) to turn into stable place cells, indicative of an ongoing functional

**Fig. 1 | Microstrokes impair spatial memory and place cell stability.**
**A** Experimental setup: Mice were trained to navigate in a linear virtual reality (VR) corridor (4 m length) during simultaneous unilateral two-photon calcium imaging in CA1 of the hippocampus. **B** Schematic illustration of microstroke induction by injection of fluorescent microspheres (20 μm diameter) into the common carotid artery, inducing microstrokes. **C** Representative two-photon images of the same field of view (FOV) in CA1 showing GCaMP6f fluorescence before (left) and after (middle) stroke induction. Damaged tissue overexposed the sensor and is shaded red. Individual neurons and their ΔF/F calcium fluorescence traces could be tracked over several weeks (right). Landmark blood vessels are marked in red. Scale bars = 25 μm. **D** Timeline showing the sequence of events of the experiment in days relative to stroke induction. Chronic two-photon imaging in CA1 was performed while animals were navigating in the VR corridor during three experimental phases: before stroke ("healthy", ≥−5 to 0 days before stroke), early (0−7 days) and late (>7 days – 28 days) after stroke. $N = 5$ independent experiments were performed with stroke and sham animals. **E** Histograms depicting the profile of an example mouse to lick for a water reward in the VR corridor. Green areas indicate reward zone locations. **F** Number of microspheres in the brain plotted against the task performance (spatial information of lick profile, in bits) during the three phases. Inset shows Pearson's correlation coefficients, data points are individual animals in each stroke phase. "Healthy" represent data points from mice before microsphere injection serving as controls. **G** Left: Like **F**, but task performance plotted against the percentage of all imaged cells identified as place cells (place cell ratio). Right: Pearson's correlation coefficients of task performance with metrics of neural activity such as the place cell ratio, the stability of neural spatial activity maps across trials (within-session stability) and the mean firing rate for the three phases. Error bars represent 95% confidence intervals. **H** Spatial activity maps of three exemplary neurons being active at the same corridor location (stable place cell), different locations (unstable place cell), or at no specific location (noncoding cell) during four healthy sessions. **I** Spatial activity maps of neurons imaged on multiple days throughout the experiment and sorted into the three functional classes from H (each row represents an individual neuron tracked before and after stroke). Boxplots are drawn with the box extending from the 25th to 75th percentiles, with the centre line at the median. Whiskers reach to the minimum and maximum values of the distribution. **J–L** Percentages of all tracked cells that were classified as stable place cells (**J**), unstable place cells (**K**) and noncoding cells (**L**). In J–L, statistics were evaluated using two-way repeated-measures ANOVA with the Greenhouse-Geisser correction and Tukey-Kramer multiple comparisons test. Asterisks indicate significances: $*p < 0.05$, $**p < 0.01$, $***p < 0.001$.

consolidation in the network (Sham group: uPC-sPC transitions: $4.2 \pm 1.6\%$, $p = 0.027$; uPC-NC transitions: $-6.9 \pm 1.8\%$, $p = 0.004$, Fig. 2D). On the long-term, stroke animals appeared to show a trend of recovering stability, with transition probabilities of stable place cells becoming similar to sham animals, although individual transitions were still not significantly different from chance level due to an increased variance (Figure S3).

### Functional stability of spatial coding influences cognitive outcome

Having identified the loss of stability of spatial coding in surviving neurons as a major effect after stroke, we next investigated how important the stability of spatial coding was for the cognitive outcome of the animals in the spatial navigation task. When comparing the post-stroke task performance to correctly identify reward zones in the VR corridor to the pre-stroke level, all stroke mice displayed a significant cognitive deficit 3 days after stroke (< 75% below their healthy baseline, Fig. 3A). However, while some animals recovered from the cognitive deficits within 10 days after stroke ("Recovery" group, Fig. 3A), this was not the case for other animals that showed a chronic cognitive deficit ("No-Recovery" group, Fig. 3A). Animals with this chronic cognitive deficit lost a significant fraction of place cells with stable place fields on long-term compared to sham animals (Stable place cells in late post-stroke: No-Recovery group: $5.5 \pm 2.8\%$ versus Sham group: $24.4 \pm 4.5\%$, $p = 0.009$, Fig. 3B). Notably, No-Recovery animals had also significantly larger lesions than Recovery mice in post-mortem histological analysis (Figure S1D). Animals with a recovery of the cognitive deficit however, showed only transiently a reduced number of stable place cells early after stroke (Recovery group: $4.5 \pm 2.1\%$ versus Sham group: $14.7 \pm 3.3\%$, p = 0.050), with the number of stable place cells increasing again, late post-stroke with no significant difference from sham animals (Recovery group: $15.1 \pm 4.8\%$, versus Sham group: $p = 0.084$). These results suggest that the maintenance of individual place cells to keep their preference for a place field is important for the cognitive outcome.

We then examined if population activity coding can predict the position of the animal in the VR corridor before and after stroke[16–18]. To investigate whether microstrokes affect the quality and stability of this encoding, we applied a Bayesian decoder to predict the most likely corridor position at each frame from neural activity of the same session[15,19] (Fig. 3C). The decoder was able to predict the correct position (Fig. 3D) and detect reward zones (Fig. 3E) significantly above chance levels in all groups in the healthy condition. However, the accuracy of the decoder and its sensitivity to predict reward zones were significantly lower for both stroke groups compared to sham early post-stroke (Decoder accuracy, Fig. 3D: Sham: $23.8 \pm 2.6\%$, Recovery: $11.8 \pm 0.7\%$, versus Sham: $p = 0.003$; No-Recovery: $9.3 \pm 1.1\%$, versus Sham: $p < 0.001$. Reward zone sensitivity, Fig. 3E: Sham: $85.2 \pm 2.4\%$, Recovery: $72.5 \pm 3.9\%$, versus Sham: $p = 0.063$; No-Recovery: $72.1 \pm 1.6\%$, versus Sham: $p = 0.002$). The decoder showed a significantly improved accuracy and sensitivity to predict the animal's position for the Recovery group in the late phase after stroke (Accuracy: late post-stroke: $21.2 \pm 2.0\%$, versus early post-stroke: $p = 0.038$; Sensitivity: late post-stroke: $83.5 \pm 1.9\%$, versus early post-stroke: $p = 0.045$), consistent with the recovery of the cognitive deficits in these mice.

To further understand the long-term stability of neural network encoding for spatial information, we adapted the Bayesian decoder by training it on the last healthy session before stroke and applying it to all other sessions. This long-term decoder could predict the corridor positions of sham mice significantly above chance level throughout the entire experiment (Fig. 3F). However, the ability of the long-term decoder to predict the position of animals based on their population activity was abolished for animals with microstrokes and particularly for those with a chronic deficit (Fig. 3F, Decoder accuracy in early post-stroke: Sham: $9.1 \pm 1.3\%$ versus No-Recovery: $2.0 \pm 0.4\%$, $p < 0.001$; late post-stroke: Sham: $6.4 \pm 0.8\%$ versus No-Recovery: $2.2 \pm 0.7\%$, $p = 0.007$).

### Stability, precision and persistence of functional network structure are markers for cognitive outcome

We next examined the stability of spatial memory on a network level. We performed population vector correlation (PVC)[15], which quantifies the stability of the spatial activity of a network across days and can be visualized in cross-correlation matrices (Fig. 4A top). Whereas sham animals showed a stable pattern of population activity relative to the position of the mice in the VR corridor throughout all stages of the experiment, animals in the No-Recovery group revealed a disturbed pattern compared to the healthy condition throughout both post-stroke phases (Fig. 4A). In contrast, the pattern of population activity recovered 2–4 weeks after stroke for animals in the Recovery group, consistent with their cognitive recovery in the spatial navigation task (Fig. 4A). Averaging the cross-correlation matrices across corridor location offsets yields summary curves that represent the similarity of neural activity at any two corridor locations with increasing distance (Fig. 4A bottom), allowing not only to quantify the stability of spatial information coding in neural populations across time, but also the spatial specificity and precision of that coding. In sham animals, the

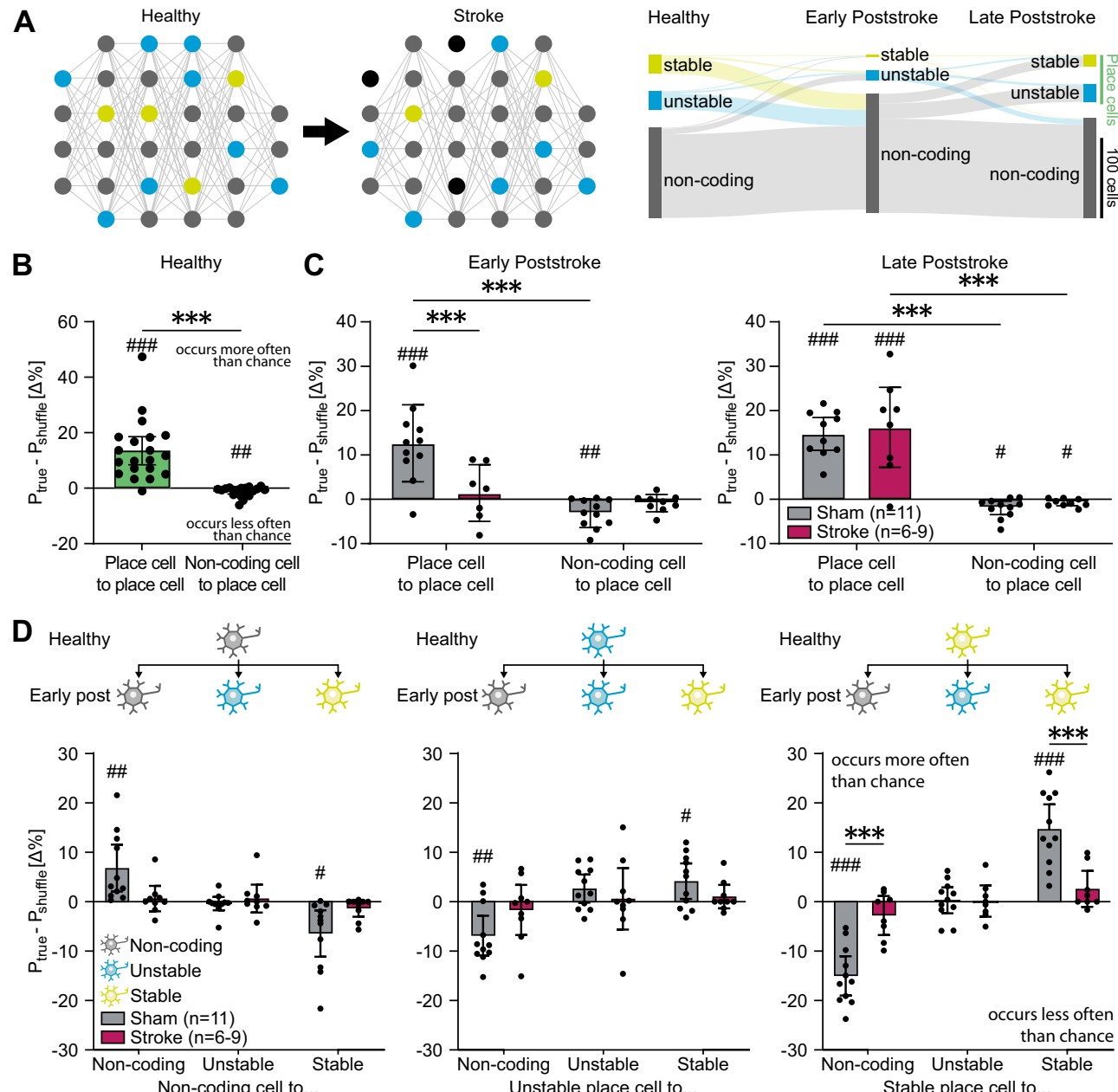

**Fig. 2 | Microstrokes induce place cell turn-over. A** Left: Scheme reflecting how neurons in hippocampal networks can transition between functional classes, which can be affected by microstrokes. Right: Sankey diagram showing the number of neurons in all stroke mice maintaining or changing functional cell classes between stroke phases: Neurons can transition to and from being non-coding for spatial information (NCs, grey), place cells (PCs, green), either stably (yellow) and unstably (blue) coding for space. Height of bars represent number of neurons in each class per phase. Connection width is the number of cells that switch functional class. **B** The probability of place cells (green) or non-coding cells (grey) to become place cells (PCs) on the next days in healthy networks, expressed as the probability difference in the true data compared to a shuffled distribution. ΔP > 0 indicates that the transition occurs more often than chance, ΔP < 0 less often than chance. *N* = 20 mice. **C** Transition probabilities of PCs (left two columns per graph) or NCs (right two columns per graph) to become PCs between experimental groups (sham and stroke) during post-stroke phases comparing data from imaging sessions every 3rd day. **D** Probabilities of non-coding cells (grey), unstable (blue) and stable (yellow) place cells to maintain or switch their functional class from the healthy to the early post-stroke phase. Data are presented as mean values +/- SD. Asterisks indicate significances of two-way repeated-measures ANOVA with Bonferroni multiple comparisons test. Hash symbols indicate significances of one-sample two-sided t-tests with Bonferroni correction to chance level (ΔP = 0). Significance levels: */# $p < 0.05$, **/## $p < 0.01$, ***/### $p < 0.001$.

matrices and curves showed clear periodic sequences corresponding to the layout of the VR corridor with repetitive reward zones and inter-reward zone areas with diverse wall patterns (Fig. 4A). In contrast, periodicity was lost in No-Recovery mice with only flat population vector curves after microstrokes (Fig. 4A), while periodicity re-emerged in animals of the Recovery group. The loss of periodicity in the No-Recovery mice did not simply reflect changes in behavior measured in lick rates as shown in Figure S2E, F and Figure S4).

The y-intercept of the population vector curves represents the correlation of the population activity at the same corridor location on two different days. It can be interpreted as a measure of the cross-session stability of the encoded memory (long-term network stability, Fig. 4A). When comparing the y-intercept between experimental groups and across time (Fig. 4B), we find that mice in the Sham and Recovery groups displayed a significant increase in cross-session stability of the neuronal network over time, suggesting a consolidation of

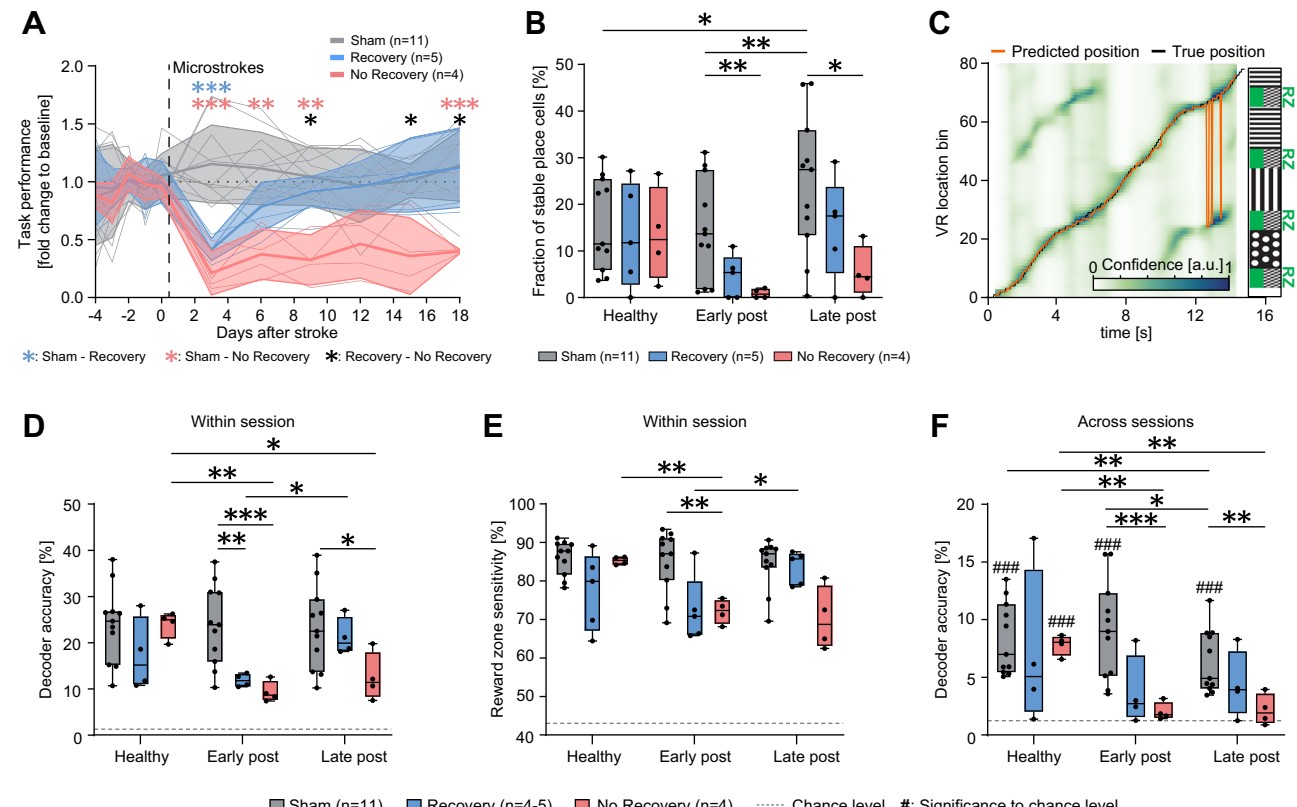

**Fig. 3 | Functional stability of spatial coding influences cognitive outcome.**
**A** VR performance after stroke relative to the healthy baseline revealing two different cognitive outcomes: A subset of mice showed a chronic cognitive deficit (No-Recovery group in red), while another group recovered from the initial cognitive decline within one week (Recovery group in blue). Sham-operated animals (grey) maintained their VR performance. A mixed-effects model with the Greenhouse-Geisser correction and Tukey-Kramer multiple comparisons test revealed significantly reduced task performances depending on the stroke outcome group ($F_{Group \times Time}$(22, 180) = 5.380, $p < 0.001$). Asterisks denote significance between sham and Recovery (blue) or No-Recovery (red) groups, and between Recovery and No-Recovery groups (black). Data are presented as mean values +/- SD. **B** Percentages of stable place cells out of all imaged neurons in the three outcome groups (Sham, $n = 11$; Recovery, $n = 5$; No-Recovery, $n = 4$) across experimental phases. **C** Predicted corridor positions generated by a Bayesian decoder (orange) from neural ΔF/F traces and true position (grey) of a single exemplary trial. The background heatmap visualizes frame-wise decoder confidence (arbitrary units) for each VR position bin. The bin with the highest confidence is the prediction of the decoder for each frame. Prediction errors (e.g. around second 13) can often be attributed to the confusion of different reward zones with similar wall patterns. **D** Accuracy (percentage of frames with correctly predicted position bin)
of the decoder indicating performance on data from the same session on which the decoder was trained (within-session decoder) in the three outcome groups (Sham, $n = 11$; Recovery, $n = 4$; No-Recovery, $n = 4$) across experimental phases. Dashed line indicates chance level (1.25%). Data from one mouse was excluded with a decoder accuracy not significantly different from the chance level in the healthy condition. **E** The sensitivity of the within-session decoder to detect reward zones (percentage of frames within reward zones correctly predicted as being in a reward zone) in the three outcome groups (Sham, $n = 11$; Recovery, $n = 4$; No-Recovery, $n = 4$) across experimental phases. Dashed line indicates chance level (43.75%). **F** Accuracy of the cross-session decoder (decoder trained on the last healthy session and applied to all other sessions) in the three outcome groups (Sham, $n = 11$; Recovery, $n = 4$; No-Recovery, $n = 4$) across the experiment. Hash symbols indicate significance (one-sample two-sided t-tests with Bonferroni correction) to chance level (dashed line, 1.25%). Boxplots are drawn with the box extending from the 25th to 75th percentiles, with the centre line at the median. Whiskers reach to the minimum and maximum values of the distribution. Group differences in **B**, **D**–**F** were evaluated with two-way repeated measures ANOVA with the Greenhouse-Geisser correction and Tukey-Kramer multiple comparisons tests. Asterisks and hash symbols indicate significances: */# $p < 0.05$, **/## $p < 0.01$, ***/### $p < 0.001$.

network activity for spatial memory (y-intercept change over time: Sham group: F(1.4,14.1) = 12.50, $p = 0.002$; Recovery group: F(2.1,8.5) = 6.94, $p = 0.016$; Fig. 4B). In contrast, animals with a chronic deficit did not show improved cross-session stability over time (No-Recovery group: F(2.1,6.4) = 1.1, p = 0.405), which instead remained on a significantly lower level late after stroke (y-intercept in late post-stroke: No-Recovery: 0.41 ± 0.03; Sham: 0.59 ± 0.03, versus No-Recovery: $p = 0.003$; Recovery: 0.64 ± 0.06, versus No-Recovery: $p = 0.030$; Fig. 4B)

Next, we quantified the ability of the neuronal networks recorded in CA1 to distinguish nearby corridor location as a measurement of the precision of spatial coding. The initial slopes of the mean PVC curves (Fig. 4A) show the rate at which population activity becomes different when comparing different locations, with a steeper slope indicating a higher spatial discrimination and memory precision. We find that

spatial discrimination is strongly disrupted in both stroke groups immediately after stroke compared to sham (precision in the healthy versus the post-stroke network activity: Sham: 1.33 ± 0.17; Recovery: 0.49 ± 0.16, versus Sham: p = 0.009; No-Recovery: 0.28 ± 0.07, versus Sham: $p < 0.001$; Fig. 4C). However, the deficits were not permanent. Although No-Recovery mice showed some improvement of spatial precision late post-stroke (early post-stroke: Sham: 1.44 ± 0.19 versus No-Recovery: 0.37 ± 0.11, $p < 0.001$; late post-stroke: Sham: 1.35 ± 0.15 versus No-Recovery: 0.79 ± 0.17, $p = 0.087$), Recovery mice displayed a faster and more complete re-establishment of spatial discrimination already within the first-week post-stroke, similar to the sham group (early: Recovery: 0.93 ± 0.23, versus Sham: $p = 0.242$; late: Recovery 1.18 ± 0.13, versus Sham: $p = 0.678$; Fig. 4C).

We also assessed the effect of microstrokes on the persistence of functional network structure in the hippocampus over time, as

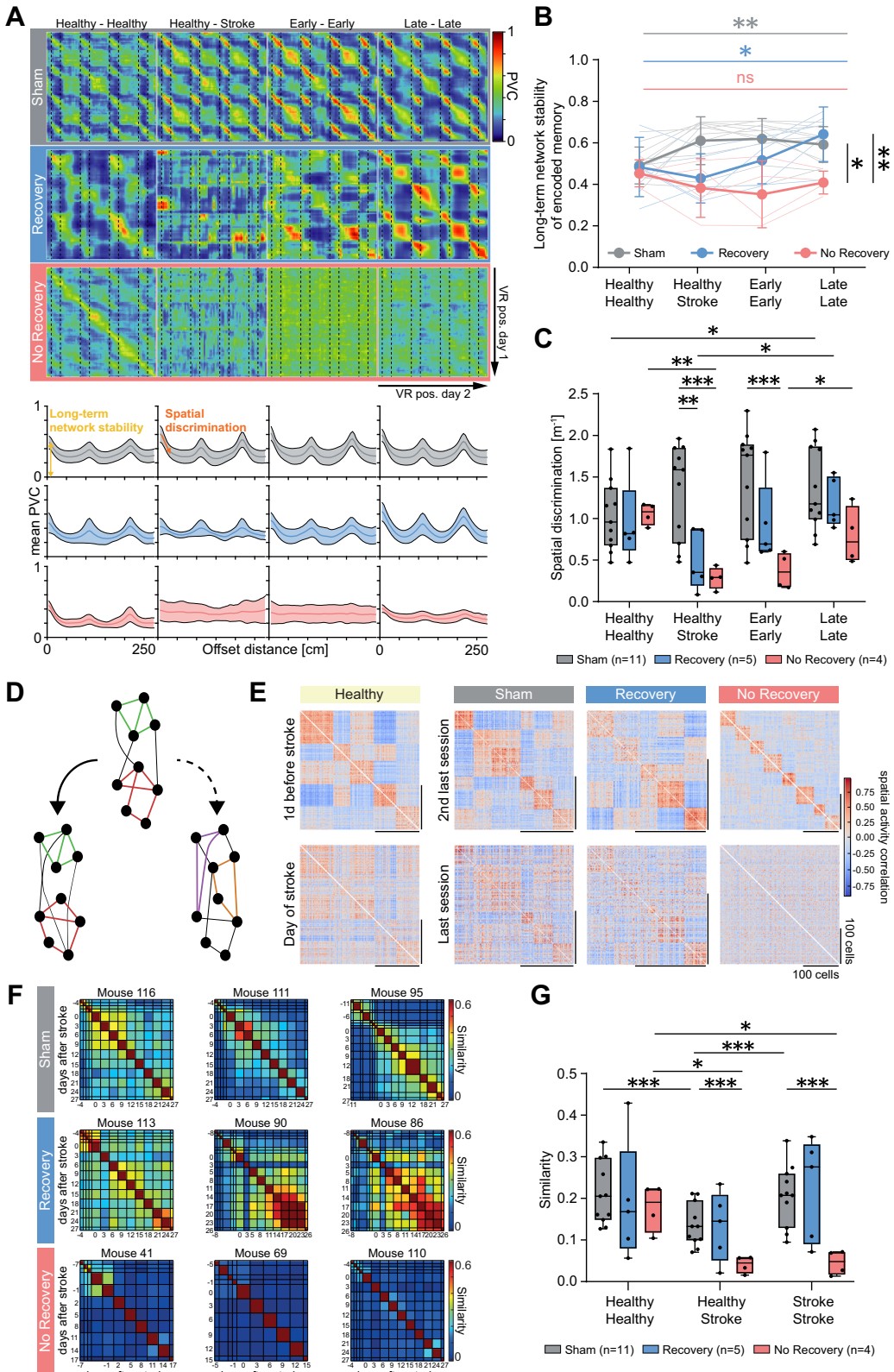

measured by the pairwise correlations of neuronal activity (Fig. 4D). First, we computed the Pearson correlations of the spatially binned activity for pairs of neurons on each day (Fig. 4E). We found that in healthy networks, the resulting pairwise correlation structure looked similar, especially across consecutive days ("Healthy", "Sham", Fig. 4E). While the same appeared to hold for Recovery mice when comparing networks after stroke ("Recovery", Fig. 4E), in No-Recovery mice no

resemblance of pairwise correlation structure could be seen after stroke. To quantify the persistence of functional networks across time in the different experimental groups, we computed the cosine similarity of the pairwise correlations of tracked neurons on all pairs of days (Fig. 4F). Indeed, while there were no differences across the three groups of mice before stroke induction, No-Recovery mice showed significantly lower pairwise correlation similarity after stroke

**Fig. 4 | Stability, precision and persistence of functional network structure are markers for cognitive outcome. A** Top: Population vector correlation (PVC) matrices of an exemplary mouse of each outcome group across session pairs within each experimental phase. "Healthy – Stroke" depicts matrices of the last healthy session in relation to the first poststroke session. Dashed lines show reward zone borders. Below: Average PVC curves summarize matrices across corridor location offsets. Correlation peaks reflect the periodic structure of the VR corridor.
**B** Y-intercept of PVC curves, which indicates cross-session stability of functional coding, in the three outcome groups (Sham, $n = 11$; Recovery, $n = 5$; No-Recovery, $n = 4$) across session pairs within each experimental phase. Thin lines represent individual mice, thick lines with error bars show mean and standard deviation. Horizontal significance bars mark time effects (group-wise one-way repeated-measures ANOVA), vertical significance bars mark differences between groups in the later phase after stroke ( > 7 days postinjection). **C** The absolute initial slope of PVC curves, which represents spatial precision in neural location coding, with higher values (steeper slopes) indicating higher precision, in the three outcome groups (Sham, $n = 11$; Recovery, $n = 5$; No-Recovery, $n = 4$) across session pairs within each experimental phase. **D** Change of functional network structure (correlations of the spatially binned activity) over time varies with effect of stroke. Schematic to represent functional network structure before (top) and after (bottom) surgery, with two possible outcomes: functional structure before stroke largely persists (left, solid arrow) or significantly changes (right, dashed arrow).
**E** Example matrices of functional correlations of spatially binned activity on subsequent experimental sessions. In healthy mice, functional structure on a given day (bottom) largely resembles the functional structure on the previous day. After stroke, sham mice and recovery mice exhibit similar functional structure of their spatial activity maps across consecutive days. Instead, No-Recovery mice show very different functional structure of their spatial activity maps even on subsequent sessions. **F** Cosine similarity of functional correlations of spatially binned activity on different days (computed for the off-diagonal elements of the spatial correlation matrices shown in **E**). **G** Mean similarity in the three outcome groups (Sham, $n = 11$; Recovery, $n = 5$; No-Recovery, $n = 4$) across session pairs within each experimental phase. Each data point shows the mean of all similarities in **F**, when the sessions being compared are both before stroke (Healthy-Healthy), before and after stroke (Healthy-Stroke) and both after stroke (Stroke-Stroke). Boxplots are drawn with the box extending from the 25th to 75th percentiles, with the centre line at the median. Whiskers reach to the minimum and maximum values of the distribution. For **A, B** data are presented as mean values +/- SD. Group differences in **B, C** and **G** were evaluated with two-way repeated measures ANOVA with the Greenhouse-Geisser correction and Tukey-Kramer multiple comparisons tests. Asterisks indicate significances: $^*p < 0.05$, $^{**}p < 0.01$, $^{***}p < 0.001$.

compared to Sham (Stroke-Stroke: No-Recovery: $0.05 \pm 0.02$, Sham: $0.21 \pm 0.02$, $p < 0.001$; Fig. 4G). We also observed a lower similarity when comparing the pairwise correlations of sham mice separated by many days (healthy - stroke) as opposed to consecutive days (healthy – healthy), consistent with a slight drift (Sham: Healthy-Stroke: $0.13 \pm 0.02$; Healthy-Healthy: $0.21 \pm 0.02$, vs. Healthy-Stroke: $p < 0.001$; Fig. 4G). The similarity between pairwise correlations of No-Recovery mice separated by many days, however, was significantly lower than this "normal" ongoing drift (Healthy-Stroke: No-Recovery: $0.04 \pm 0.01$, vs. Sham: $p < 0.001$; Fig. 4G).

Hence, in healthy mice, groups of neurons show a pairwise correlation structure of their spatially binned activity that persists across time, with some drift, demonstrating the stability of functional subnetworks. Following stroke, stable functional subnetworks return only in recovery mice (albeit different than before stroke), while in No-Recovery mice, functional subnetworks continue changing across time, consistent with the circuit's inability to recover normal behavior. Therefore, these data suggest that the stability of functional networks over time could be essential in maintaining cognitive capability after microstrokes.

### Synchronous activity close to salient locations in animals with recovery of memory deficits

To investigate a functional relationship between individual surviving neurons after stroke, we quantified the synchronicity of pairs of neurons by correlating their ΔF/F traces (Fig. 5A). The average ΔF/F correlation of the whole network calculating the mean correlation between all pairs of neurons per network did not reveal a significant difference between experimental groups after stroke (Fig. 5B). However, we identified a different composition of functional cell classes, when comparing activity of neuronal pairs with high and low correlated activity. When plotting the cumulative distribution of correlated firing activity of place cell and non-coding cell pairs, we examined a clear difference for the 95th percentile of neuronal pairs (with highest pairwise firing) between place cells and non-coding cells in the healthy condition (Fig. 5C). Most cell pairs with high synchronous activity (the 95th percentile at the dashed red line) were place cells, justifying a subsequent quantification of the composition of functional cell types in this 95th percentile subset of highly correlated cell pairs. We then examined how the composition of this 95th percentile changed after stroke. We found in animals of the No-Recovery group, a reduced percentage of synchronous active place cells early after insult (Sham: $18.4 \pm 2.1\%$ vs. No-Recovery: $8.3 \pm 2.3\%$, $p = 0.048$; Fig. 5D), while

animals with a recovery from the cognitive deficits (Recovery group) revealed an increase of the percentage of synchronous active place cells after stroke (Early post-stroke: $15.9 \pm 3.0\%$ vs. Late post-stroke: $24.5 \pm 3.6\%$, $p = 0.025$, Fig. 5D). Accordingly, the percentage of non-coding pairs of neurons with highly correlated activity was higher in animals with a chronic cognitive deficit compared to sham animals after stroke (Early post-stroke: Sham: $4.4 \pm 0.1\%$ vs. No-Recovery: $4.9 \pm 0.1\%$, $p = 0.015$; Late post-stroke: Sham: $4.3 \pm 0.2\%$ vs. No-Recovery: $4.8 \pm 0.1\%$, $p = 0.033$; Figure S5D).

We next asked if neurons with highly correlated activity also share the same spatial representation. Correspondingly to the results in Fig. 5D we also found a decline of spatial correlation in animals of the No-Recovery group early after stroke (Fig. 5E). We then examined whether these highly correlated functional cells were clustered around salient locations such as reward zones. Although no difference of place field location was found in the healthy situation among the three groups (Fig. 5G), sham animals had significantly more place fields in the reward zones than at less salient locations (far: $24.6 \pm 3.1\%$, in: $44.5 \pm 6.1\%$, $p = 0.029$; Fig. 5G). Late after stroke, animals of the Recovery group revealed a significant increase of place cell pairs with place fields close to reward zones (close: $58.8 \pm 8.8\%$; in: $25.2 \pm 6.9\%$, vs. close: $p = 0.003$; far: $16.1 \pm 2.9\%$, vs. close: $p < 0.001$; Fig. 5G), indicative for a re-organization of place fields close to salient locations in animals with a good outcome, which was not detectable in mice with a chronic spatial memory deficit or Sham mice (Fig. 5G).

We also examined to which extent cross-day stability of place cells is driven by cells active near reward zones. We first analysed the average distance to the next reward zone of stable and unstable place cells, split by experimental group and stage. Plotting these data as a distance difference, with positive values indicating that unstable place cells are closer to reward zones, and negative values that stable place cells are closer to reward zones suggested no connection between place cell stability and distance to reward, as all datasets were not significantly different from 0 (Figure S6A). We also correlated place field distance to the next reward zone to the cross-session stability of place cells across experimental groups and stages (Figure S6B). While we found no significant correlation for animals with a stroke, sham mice displayed weak but significant negative correlation between cross-session stability and place field distance during early and late post-stroke sessions.

We also analysed if neurons with highly correlated activity are physically located close to each other in the rewiring network after stroke (Figure S5A). We found no strong relationship between

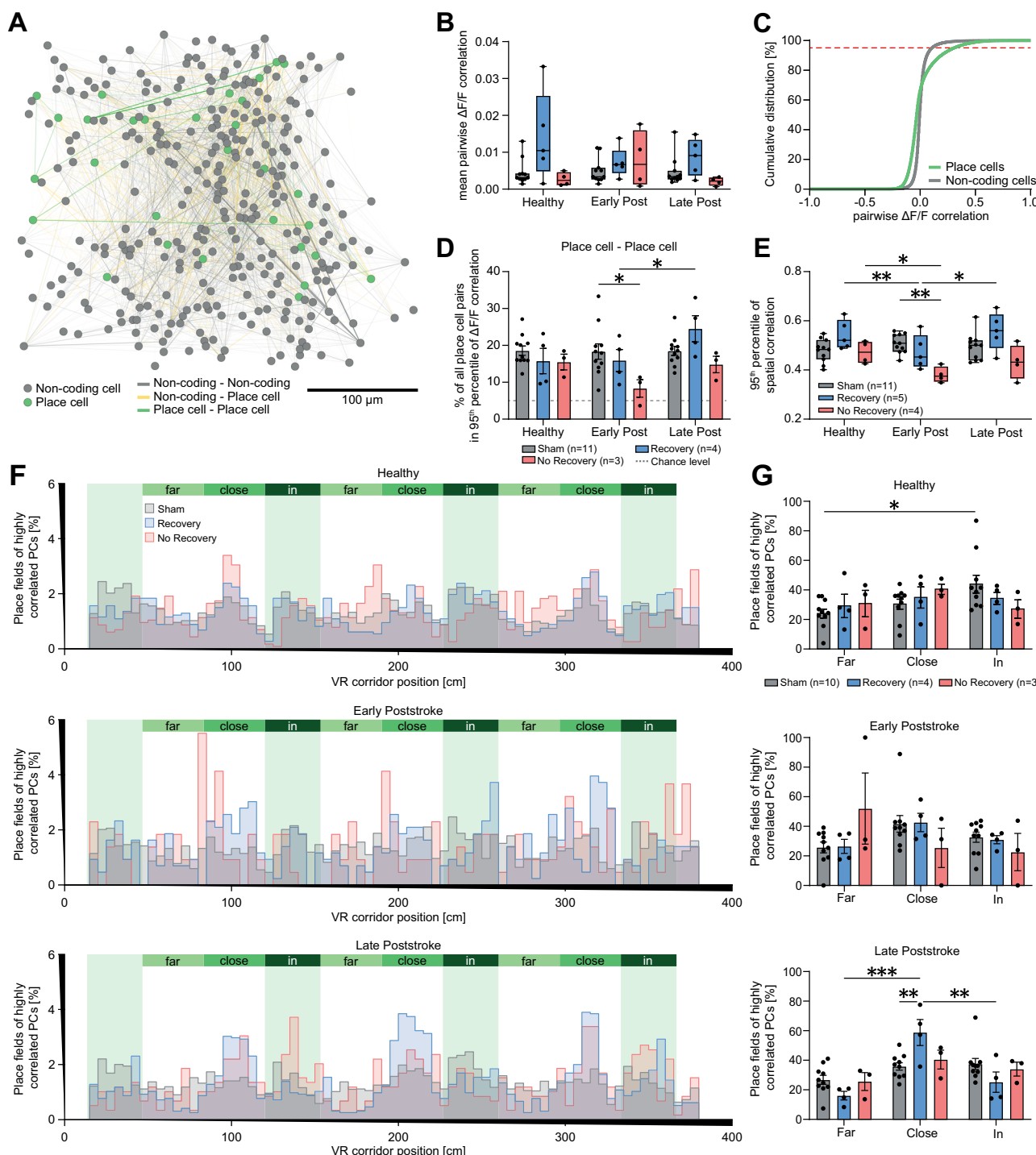

synchronous active cells and their (Euclidean) distance to each other in the chronically imaged networks.

## Discussion

We demonstrated tracking of individual neurons in the same region of interest in the hippocampus for several weeks before and after induction of microstrokes in the brain in relation to spatial memory. We used chronic 2-photon calcium imaging in mice navigating in a virtual reality corridor to dissect the functional roles of neurons in CA1 for distinct spatial information in the healthy condition and after microstrokes, and identified place cells with stable activity for a distinct place field over several days versus unstable, remapping place cells and non-coding cells for space. Furthermore, our approach

allowed us to measure the loss of spatial memory: The extent of cognitive decline and destruction of the neuronal network architecture in the hippocampus strongly correlated with the number of brain-wide microstrokes identified histologically post-mortem, suggesting a dose-dependent effect, as mice with the highest microsphere load showed the largest chronic cognitive deficits. Supporting this idea, the No-Recovery group included animals with some degree of cognitive improvement (although not to the level observed in the Recovery group). These animals had the highest fraction of stable place cells (Fig. 3B), the best decoder performance (Fig. 3D-F) in the late post-stroke period, but also the lowest number of microspheres in the brain (Figure S1C) compared to the other animals of the No-Recovery group, suggesting that our experimental model provided a continuum of

**Fig. 5 | Synchronous activity close to salient locations in animals with recovery of memory deficits. A** Network schematic showing spatial distribution and correlation of pairwise firing (Pearson's correlation of ΔF/F traces) between all imaged place cells (PCs, green) and non-coding cells (NCs, gray) in an exemplary field of view. Highly correlated neurons (95th percentile) are connected by lines. Line width and opacity represent connectivity strength, and colors identify functional coding of cell pairs (gray: NC – NC pairs; yellow: NC – PC pairs; green: PC – PC pairs). **B** Mean correlation of pairwise activity between all imaged neurons in the three outcome groups (Sham, $n = 11$; Recovery, $n = 5$; No-Recovery, $n = 4$) across experimental phases. **C** Cumulative distribution function (CDF) of pairwise ΔF/F activity between place cell and non-coding cell pairs in the healthy condition. Red dashed line indicates the 95th percentile, at which the difference between place cells and non-coding cells is largest. **D** Percentage of place cell pairs in the 95th pool of cells with highly correlated activity shows a loss of place cells pairs in the No-Recovery group. Error bars depict standard error. Dashed lines represent chance level of a uniform distribution (5%). **E** 95th percentile of correlation coefficients of spatial activity maps between all imaged neurons. **F** Histograms of place field distributions of highly correlated place cells in the three different outcome groups (Sham, Recovery, No-Recovery) and the different time points (healthy, early and late after stroke). Histograms show how many place fields were located far from, close to, or in the reward zones of the virtual reality corridor. **G** Analysis of the histograms in **F** reveal that initially, highly correlated place cells in sham mice are in particular found in reward zones, while in the late phase after stroke, animals of the Recovery group show a significant concentration of place fields close to reward zones compared to other corridor regions and compared to sham. Bar plots in D and G are presented as mean values +/- SEM. Boxplots are drawn with the box extending from the 25th to 75th percentiles, with the centre line at the median. Whiskers reach to the minimum and maximum values of the distribution. Group differences were evaluated with two-way repeated measures ANOVA with the Greenhouse-Geisser correction and Tukey-Kramer multiple comparisons test. Asterisks indicate significances: *$p < 0.05$, **$p < 0.01$, ***$p < 0.001$.

stroke severity and recovery potential, which is correlated to the total microstroke burden in the brain.

As we found only 14% of lesions affecting directly the hippocampus and most microstrokes were remote from the recording site in the hippocampus, we not only observed a strong deficit in our spatial navigation task, but in particular an impaired stability of spatial coding of individual neurons as well as affected stability of the population activity and the persistence of the functional network structure. These results suggest that local damage to the hippocampus is not necessary to impact hippocampal function. Instead, systemic reactions to widespread microstrokes such as inflammation or network remodeling[20,21] in form of altered input patterns might be sufficient to modify global brain activity, impairing neuronal networks remote to the acute lesion locations.

While chronic recordings of the same field of view in the hippocampus have been already reported in a few studies in the healthy condition[12,22] and in mice after induction of epilepsy[15], schizophrenia[23] and hippocampal lesions, our study shows that chronically monitoring the activity of individual cells and the same neuronal populations in the healthy condition and during several weeks after brain injury is possible. Although functional coding of individual neurons in CA1 of the hippocampus seems relatively labile in contrast to synchronously firing groups of neurons or neurons in other regions of the hippocampus[12], we identified stable place cells which maintained their place field activity not only during the healthy condition, but also several weeks after brain-wide microstrokes. In addition, when monitoring the same neurons in sham animals for >5 weeks we found that neurons maintained their functional class (stable or unstable place cell or non-coding cell) long-term (Fig. 2C, D and Figure S3), indicative for a consolidation of the functional network in sham animals (Fig. 4). This is in line with preliminary results from Vaidya et al. [24] claiming that there are two place cell pools – a transient and a sustained one – and that initially formed unstable, transient place cells are replaced by stable, sustained place cells over time. This consolidation might be mediated by behavioral timescale synaptic plasticity (BTSP), a recently discovered non-Hebbian mechanism where synaptic weights can be modulated by a single event, which can be temporarily separated from the synaptic input by several seconds[25–28] and which is also thought to induce place field formation in CA1[27].

We found that in healthy animals with expert knowledge in the spatial navigation task, place cells had a significantly higher probability to stay place cells than turning into non-coding cells. Strokes destroyed the functional determination of neurons within networks (Fig. 2C, D): Instead of being pre-tuned to a distinct functional class for the spatial memory task, neurons were randomly assigned to different functions within the rewiring network after stroke. A similar effect as observed in this study, has been described in animal models of schizophrenia[23] and epilepsy[15]. Thus, this return to a more disordered,

plastic state might not be unique to acute lesions, but also occur in chronic disruptions of hippocampal networks. It therefore may not be a purely detrimental process. Instead, it may be a cellular mechanism of increased homeostatic plasticity, which is commonly detected in the subacute phase of stroke. Besides the already known plasticity processes after brain injury such as macroanatomical map shifts[29–32], synaptic and dendritic turn-over[33], and molecular modifications[34,35], the loss of functional cellular identity and the promotion of neurons to new functions after stroke could be an important cellular mechanism to better adapt to the injury and allow functional rewiring of neuronal networks after brain injury such as strokes.

While on an individual cellular level, the loss of stable coding place cells influenced the outcome (Fig. 3B), on a population level microstrokes impaired location encoding (Fig. 3D-F), spatial discrimination and precision (Fig. 4), persistence of functional network structure (Fig. 4) and synchronicity of surviving place cells (Fig. 5C). All these parameters remained significantly affected in animals with a chronic deficit, while in animals with a recovery from the initial memory loss, these parameters restituted to the same level as in sham animals, highlighting these parameters as sensitive functional markers for outcome prediction and as important indicators for future interventional studies examining the positive or detrimental effects for pharmacological or rehabilitative approaches.

Finally, we found that in animals with a chronic cognitive deficit the number of highly synchronous active cells was significantly reduced. An in-depth analysis revealed that animals in the No-Recovery group particularly lose synchronously active place cells (Fig. 5C) and display nearly complete reorganization of functional connectivity networks (Fig. 4G), which was not the case for the other groups. In addition, in Recovery animals, place fields of synchronous active surviving place cells were more often located close to salient locations (Fig. 5F, G). A clustering of place fields around reward zones has been reported repeatedly[22,36–38]. However, in our experiment with chronic imaging before and after microstroke induction we found that place fields of neurons with highly correlated activity were uniformly distributed in the corridor, except for animals of the Recovery group during the later post-stroke state (Fig. 5F, G). The increased number of place fields close, but not in reward zones (Fig. 5G), may be related to the phenomenon of anticipatory licking (Figure S7B) and the reward prediction mechanism of the dopaminergic system, where the reward response is triggered by an associated stimulus (such as the visual cue of the approaching distinctive wall patterns of the reward zone).

We also examined if stability of place cells is driven by neuronal activity near reward zones as previous studies[22] revealed that place cells near reward zones have higher cross-day stability. In our data we found only weak evidence for a higher stability of place cells close to reward zones. A possible explanation may be that in our experiments mice were well accustomed to the corridor, as they had been exposed

to the same context for up to 2 months prior. In contrast, most studies use novel or changing corridors, and context exposure is often limited to a few days or weeks. It is possible that the relationship between place field reward proximity and cross-session stability is more pronounced during the early learning phase of the corridor, but the bias is slowly replaced during learning by a more uniform distribution, a phenomenon already suggested on shorter time scales by Grosmark et al.[22]. Future experiments with our setup may include imaging sessions during the learning period, which might provide further information about the temporal dynamics of these mechanisms.

Our results indicate a major protective mechanism for the recovery and preservation of spatial memory after brain injury: The survival of place cells which are able to maintain their place field preference for important information (e.g. salient locations such as reward zones) over time is predictive for a good cognitive outcome after stroke. Importantly, the synchronous activity of place cells helps them to survive and to re-stabilize the rewiring network after stroke. Thus, we show here that the well-known concept of Hebbian learning of "neurons that fire together wire together" applies also in a rewiring network after brain injury[35]. Elucidating this concept in the light of neuronal repair on a cellular resolution level after stroke has also implications for the development of novel pharmacological and stimulation strategies, which can induce the co-activation of neurons in networks to stabilize and enhance reorganizing circuits in the brain after stroke.

## Limitations

With the here presented microstroke model we aimed at mimicking key features of small vessel disease, which comprises the majority of cases of vascular dementia[39]. This includes the induction of disseminated brain-wide microstrokes, a measurable cognitive deficit and an effect on hippocampal networks. However, while in the human presentation of small vessel disease microstrokes are predominantly located subcortically, we found in our stroke model widely distributed microstrokes, in particular also in cortical regions. As bilateral microsphere injections in the common carotid arteries in mice lead in most cases to fatal outcomes[40], we chose unilateral injections of microspheres in the left common carotid artery leading to microstrokes predominantly in the left hemisphere, where we imaged the effect of the disseminated microstrokes on left hippocampal networks. However, future studies should investigate the effects of the microstrokes on the contralateral hippocampus using multi-area imaging and tracing technology to reveal structural changes of newly rewired bilateral hippocampal circuitry in relation to the cognitive outcome.

## Method

### Animals

We used adult mice aged 5–7 months at the first day of experiments of both sexes ($n = 5$ males, $n = 20$ females). $N = 9$ mice were C57BL/6 wild-type (Charles River, Germany) animals, $n = 15$ were GP5.17 transgenic mice (Jackson Laboratory, RRID: IMSR\_ JAX:025393Dana et al.[41]), and one was a triple-transgenic mouse acquired by crossing animals from the lines Snap25-IRES2-Cre-D (Jackson Laboratory, RRID: IMSR \_JAX:023525), CaMKII-tTA (Jackson Laboratory, RRID: IMSR \_JAX:007004) and Ai93D (Jackson Laboratory, RRID: IMSR \_JAX:024103). Both transgenic mouse lines expressed the calcium indicator GCaMP6f[42] in CA1 pyramidal neurons of the hippocampus. For detailed information of animals used in the experiments please refer to Supplementary Table S2. Male and female mice did not display differences in behavioural, neural or histological markers (Figure S8). However this study was also not designed to investigate sex-specific responses to microstrokes. Mice displayed strong and uniform GCaMP6f expression in the CA1, with GP5.17 mice more readily available due to faster breeding. Mice were housed in groups of two to four under a constant 12-hour dark/light cycle, constant room temperature ($22 \pm 1\,°C$) and with food and water ad libitum in standard cages ($530\ cm^2$ floor area, 7.4 L). After starting training in the virtual reality corridor, mice were transferred to larger cages ($1800\ cm^2$ floor area, 51 L) equipped with a running wheel to enhance fitness and treadmill motivation. All experiments were carried out during the active (dark) cycle of the animals and according to the guidelines of the Federal Veterinary Office of Switzerland and the license ZH241/2018 approved by the Cantonal Veterinary Office in Zurich, Switzerland. They are in accordance with the Stroke Therapy Academic Industry Roundtable (STAIR) criteria for preclinical stroke investigations[43]. The samples size for the different experimental groups was estimated by means and variance of measured data in related work[30,31,44] and predicted to be sufficient to detect a statistically significant result in ANOVA with $p < 0.05$ and power >0.8.

### Surgeries

For all surgical procedures except microsphere injections, mice were deeply anesthetized with 4% Isoflurane (700-800 mL $O_2$ flow rate). 20–30 min prior to surgery Carprofen (5 mg/kg body weight subcutaneous (s.c.)) was administered, vitamin A crème (Bausch & Lomb) was applied to both eyes, and body temperature was maintained at $36.5\,°C$ via a heating pad. Animals were fixed in a stereotaxic frame (Kopf Instruments) under 2% Isoflurane for surgical procedures. After surgery, mice were kept on a heating pad until being fully awake again and moving in the cage. Post-surgical pain was managed by Carprofen injections (s.c.) every 12 h for 1 day, and every 24 h for an additional 2 days.

### Viral injections

To induce expression of the calcium indicator GCaMP6f in wild-type mice, 300 nL of AAV9-hSyn::GCaMP6f ($1 \times 10^{13}$ vg/mL; Addgene, catalog # 100837-AAV9) were stereotaxically injected into the hippocampus (left CA1 at -2 mm AP, -1.3 mm ML) at least 4 weeks before the first session of two-photon calcium imaging. Injections were performed using a glass micropipette (intraMark, Blaubrand, 10–20 μm tip diameter) connected to a syringe and a stereotaxic micromanipulator (Kopf Instruments). After removing the scalp and drilling a hole above the injection site, the pipette was gradually inserted to the intended depth (-1.5 mm from skull surface), and the virus was injected with a flow rate of 0.1 μL/min. The pipette remained in place for 10 min post-injection to facilitate viral diffusion and reduce backflow before slow withdrawal.

### Chronic hippocampal imaging preparation

For performing chronic two-photon calcium imaging, we implanted a chronic cannula implant above the left CA1 region of the hippocampus 2–3 weeks after viral injection of GCaMP6f, according to published protocols[45–47]. After scalp removal, iBond (iBond Total Etch, Kulzer) was applied to the cleaned skull, followed by a 3 mm craniotomy and inserting a biopsy punch above CA1 (-2 mm AP, -1.5 mm ML relative to bregma, 1.3 mm depth from skull surface). The punch was withdrawn after 10 min, and the severed tissue was slowly aspirated using a blunt 22 G needle while irrigating with saline, until the corpus callosum was fully exposed. After bleeding was stopped with hemostatic sponge, a custom stainless steel cannula (3 mm diameter, 1.3 mm length) sealed with a coverslip (3 mm diameter, 0.17 mm thickness) via UV-cured dental cement (Tetric EvoFlow A1, Ivoclar Vivadent) was inserted into the brain to cover the corpus callosum and fixed to the skull with dental cement. In addition to the pain medication (Carprofen), mice received the antibiotic Baytril (10 mg/kg s.c., Bayer) for 2 days. 1 week after cannula insertion, a custom aluminium headpost (10 mm inner diameter, 12.3 mm outer diameter, 1 mm thickness) was centered on the hippocampal window and attached to the cement with a ceramic composite (Charisma, Kulzer) and additional dental cement. After UV curation, the inside wall of the cement was coated with black nail

polish to minimize imaging noise. After a recovery period of 3 days, handling and behavioral training started.

## Intraarterial microsphere injection

To induce hemisphere-wide microlesions, we injected red-fluorescent 20-μm diameter PMMA microbeads (Poly-An, Cat. No. 19096-2) into the left common carotid artery (CCA) of the animals[48,49]. Mice were deeply anesthetized with Medetomidine (250 μg/kg, Domitor, Orion Pharma), Midazolam (5 mg/kg, Dormicum, Roche) and Fentanyl (50 μg/kg, Sintetica), and positioned on their back on a heating pad. A midline incision above the thyroid gland exposed the CCA, into which 10–20 μL of the 1% microsphere stock solution, diluted in 180 μL saline was injected with a 33 G needle. The external carotid artery was transiently ligated during the injection with surgical thread, and CCA flow was slightly restricted during the injection (30 s) to minimize bleeding. After removing the needle, pressure on the injection site was applied with fine cotton swabs until bleeding ceased (6–20 min). The CCA was sealed with tissue adhesive (Vetbond, 3 M), the incision was sutured, and the animal was placed on a heating pad until fully awake. Sham-operated mice underwent identical surgery, without the microsphere injection. After a 2-day recovery, experiments resumed on the third day.

## Virtual environment and task

We built a custom-made virtual reality (VR) setup, optimized to fit under a standard commercial two-photon microscope (HyperScope, Scientifica Ltd.). Animals could navigate in the virtual reality by walking on a custom treadmill consisting of a 5 cm wide black velvet ribbon belt stretched over two 10.5-cm diameter plastic wheels. The belt rested on a Teflon bar, with a smooth underside to reduce friction and a felted surface for improved grip. An optical rotary encoder (1440 pulses/rotation, Phidgets, Cat. No. 3530\_1) captured back wheel movements, controlling the VR displayed on three TFT-LCD monitors (10.1", 1366×738, LG, Cat. No. LP101WH1) positioned around the animal in 11–17 cm distance to its eyes. The virtual environment created with Unity (v2018) consisted of a linear corridor featuring four distinctly patterned sections divided by reward zones (RZs) marked by salient visual and auditory (200 ms, 8 kHz, 60 dB) cues. RZs were 40 cm wide and centered at 30, 137, 243 and 350 cm in the standard 4 m corridor. We chose a multisensory environment similar to published designs[46] to improve the learning rate and spatial coding in hippocampal CA1. Mice had to report RZs to receive water rewards (10 μL, controlled by a solenoid valve (SMC, Cat. No. VDW22JA)) by touching the metal spout with their tongue, where a capacitance sensor detected these touches as licks. When the animal reached the end of the corridor, it was virtually reset to the start of the corridor during a 1 s screen blackout. The VR was controlled by a Python 3.7 script, while a LabView2014 program orchestrated data acquisition and flow between the sensors, microscope and VR.

## Training protocol

At least three days after head-post implantation, the drinking water of the mice was replaced with 2% citric acid water. This method was shown to induce thirst and motivation for water rewards without restricting home-cage water access[50], while the body weight was controlled to be kept at >85% of the pre-experimental value. After familiarizing the mice with handling and head fixation over 2–4 days, they underwent training to selectively lick in designated reward zones (RZs) through 15–30-min daily sessions. Initially, passive water rewards were given regardless of the RZ location after running 20–30 cm. After two to three sessions, passive rewards were restricted to RZ locations in a 170 cm corridor. Once mice were running consistently, the corridor length was extended to 400 cm by adjusting the gain of the rotary encoder, as well as introducing active rewards requiring licking within an RZ to receive water. Animals were not punished for licking behavior outside RZs.

As the strategies how to identify reward zones in the spatial navigation task varied among animals and six animals (IDs 33, 69, 91, 93, 110 and 122, see Table S2) showed extensive anticipatory licking in front of reward zones (Figure S7B) also during expert phases, we used the spatial information (SI) content of the lick histogram. The SI-based performance metric also better reflected the focused licking patterns of such strategies, while simpler metrices such as the percentage of licks within reward zones (lick hits) could not accurately represent the performance of the animal. To quantify the spatial information (SI) content, we first created a lick histogram by binning licks (defined as the time points when the tongue of the animal touched the water spout) into 120 spatial bins, and computing the lick probability (the fraction of trials with at least one lick) for each position bin (Fig. 1E). The SI content of this lick histogram was then determined using a formula commonly applied to measure neural spatial information content[15,51]:

$$SI = \sum_{i=1}^{N} p_i \frac{\lambda_i}{\bar{\lambda}} \log_2 \frac{\lambda_i}{\bar{\lambda}} ; p_i = \frac{t_i}{\sum_{i=1}^{N} t_i} ; \bar{\lambda} = \sum_{i=1}^{N} p_i \lambda_i \qquad (1)$$

where $t_i$ represents the occupancy time spent in the $i$-th bin and $\lambda_i$ represents the lick probability in the $i$-th bin. Animals were considered well-trained once they consistently performed at a plateau level for three consecutive days, and imaging sessions began. Animals required 13–22 days to learn the task, after which baseline neuronal activity in the CA1 was recorded across five days using two-photon calcium imaging. Afterwards animals were randomly assigned to the sham and stroke group. Microstrokes were induced after the last baseline imaging session on the same day (day 0), followed by post-stroke imaging sessions every third day up to 4 weeks after stroke to study the "pure" effect of the disseminated lesions on the hippocampal network and to avoid neurocognitive training/rehabilitation induced plasticity. Animals were not exposed to the corridor outside of imaging sessions.

## Experimental stages & groups

To capture temporal developments of behavioral and neural metrics in relation to the microsphere injections, we structured the experiment into three phases: "healthy" (before injection), "early post-stroke" (≤ 7 days after injection), and "late post-stroke" (> 7 days–28 days after injection). The average standard deviation of task performance for all animals across healthy sessions was computed to be 23.8%. Rounded to 25%, a threshold of 75% was defined, where mice that received microsphere injections and performed below 75% of their prestroke VR performance in both early and late post-stroke phases were labeled "No-Recovery", whereas animals with an average relative VR performance of ≤75% only in the early, not late post-stroke phase, were categorized as "Recovery" (Figure S7A). All mice with an initial performance deficit of ≤75% after microsphere injections were grouped as "Stroke". Mice that maintained >75% relative VR performance in both post-stroke phases after microsphere injections and did not have significantly more detected spheres than sham-operated mice in their brains were pooled with sham-operated mice for subsequent analysis (Figure S1C). $N = 5$ independent experiments were performed with stroke and sham animals and data were pooled from these experiments as results from independent experiments did not statistically differ from each other.

## Two-photon imaging

Chronic two-photon calcium imaging of mice navigating in the VR corridor was performed with a standard commercial two-photon Galvo-Resonant scanning imaging system (HyperScope, Scientifica Ltd.) controlled by ScanImage v2017b (Vidrio Technologies[52]). A 920-nm laser beam from a tunable Ti:Sapphire laser (Mai Tai BB, Spectra-Physics) was targeted onto CA1 pyramidal neurons through a 16× water-immersion objective (0.8 NA, Nikon). Emitted fluorescence was detected by a

GaAsP photomultiplier tube (Scientifica Ltd.) after passing through a 525/50 nm band-pass filter. Beam power under the objective was adapted to GCaMP6f expression levels to 25-60 mW. The emission light path between the focal plane and the objective was shielded with a 3D-printed plastic cylinder fixed between the objective and the head-post, as well as black nail polish on the dental cement of the window pre-paration, to reduce light contamination from the VR monitors. Images of 512×512 pixels, corresponding to a field of view (FOV) of 830×830 μm, were acquired at a frame rate of 30 Hz. Prior to the experiment, a net-work with strong GCaMP6f expression encompassing a maximal num-ber of pyramidal cells, was chosen for each mouse, and the FOV was manually aligned with this network on consecutive imaging sessions to monitor as many neurons as possible over time. Pre-processing of all two-photon imaging data was performed with CaImAn[53], utilizing con-strained non-negative matrix factorization to perform piecewise-rigid motion correction, functional region of interest (ROI) extraction, background correction and quality evaluation. CaImAn parameters were adapted for each mouse to detect the maximum number of active cells while correctly rejecting non-neuronal ROIs, and were kept con-stant across the experiment. The resulting fluorescence traces were detrended with CaImAn using the following formula:

$$\Delta F/F = \frac{F - F_0}{B_0 + F_0} \quad (2)$$

where $F$ is the background-corrected fluorescence trace of the neuronal ROI, and $B_0$ and $F_0$ are the baseline fluorescence traces of the background and ROI respectively. The baseline of each trace is set at the percentile of the mode of the data, which was computed for each trace over a sliding window of 1000 frames using a diffusion kernel density estimator[53,54], which yielded baseline percentiles of 45.4 ± 5.5 (mean ± standard deviation) across all ROIs. Finally, the resulting ΔF/F traces were transformed for spike rate analysis into deconvolved spike probabilities with the CASCADE algorithm[55].

### Single-cell tracking

Aligning the FOV to the same network in subsequent imaging sessions enabled us to semi-automatically match individual neuronal ROIs across the experimental timeline. To identify the same cells across sessions, we computed the shifts between both FOVs using non-rigid translation, accounting for non-rigid changes in the underlying tissue, particularly post-injection, occurring over weeks. Each FOV was split into four patches, estimating the subpixel translation shift for each patch with scikit-image's (v0.19.2) phase cross-correlation using fast Fourier transform[56,57], and upscaling single-patch shifts via spline interpolation to generate pixel-wise shifts. Putative matched cells were first selected by identifying the nearest neighbor for each neuron through a k-d tree (Scipy v1.10.0[58]), then visually inspected and curated in a custom-built interactive web application developed with Dash (v2.0.0, Plotly Technologies). Only ROIs that were manually confirmed to be the same neuron were used for single-cell analysis.

### Neuronal activity analysis

**Linear corridor analysis.** Place cell classification was performed according to published criteria[45,46]. First, ΔF/F traces and occupancy times were spatially binned using 5 cm wide bins and a running velocity threshold of 5 cm/s to yield spatial activity maps for each neuron. After smoothing with a Gaussian kernel (σ = 5 cm), spatial activity maps were screened for putative place fields, which were defined as locations with a ΔF/F value above 25% of the difference between the maximum and baseline ΔF/F (average ΔF/F of the lower quartile activity) of this trace. Putative place fields also had to pass three criteria: (1) the place field had to have a minimum width of 15 cm, (2) the mean ΔF/F inside the field had to be 6x higher than outside the field, and (3) significant transients had to be present for at least 20% of the time the animal

moved inside the field. Significant transients were periods of at least 0.5 seconds in the unbinned ΔF/F trace with fluorescence above 3 σ (noise level σ estimated from FWHM of the ΔF/F distribution). The significance of the place field $p_{pf}$ was estimated using bootstrapping[45]. The ΔF/F trace was split into 50 frames long pieces and randomly shuffled, and place cell detection was performed on the shuffled trace. This process was repeated 1000 times, and $p_{pf}$ was defined as the fraction of shuffles where a place field passed all three criteria. Cells were classified as a place cell if their spatial activity maps contained at least one place field that passed all three criteria and had a significance of $p_{pf} \le 0.05$.

The within-session stability of the spatial activity map of each neuron was determined by computing the Pearson correlation coeffi-cient between the first and second halves of the trials, as well as between odd and even trials. The two coefficients were Fisher Z-transformed and averaged to derive a single stability measure.

To quantify cross-session stability of place cells, we computed the Pearson correlation coefficient between the spatial activity maps of all session pairs separated by 3 days across all tracked cells, and the Fisher transformed coefficients were averaged across all sessions within each period, yielding a stability score per phase for each neuron. The baseline stability score for a network was established as the median prestroke stability score across all neurons within that network. Sub-sequently, each place cell was categorized as "unstable" or "stable" for the early and late post-stroke phases based on whether its stability score for that phase was lower or higher, respectively, than baseline stability score of its network.

**Population vector correlation.** The population vector correlation (PVC) for a population of N neurons between two corridor positions $x$ and $y$ was defined as:

$$PVC(x,y) = \frac{\sum_{j}^{N} \lambda_j^1(x)\lambda_j^2(y)}{\sqrt{\left(\sum_{j}^{N} \lambda_j^1(x)\lambda_j^1(x)\right)\left(\sum_{j}^{N} \lambda_j^2(y)\lambda_j^2(y)\right)}} \quad (3)$$

where $\lambda_j^1$ and $\lambda_j^2$ are the spatial activity maps of neuron $j$ in two different sessions. Full PVC curves were created by averaging the PVC values across all corridor positions for position pairs with a $\triangle x = |x - y|$ location offset ranging from 0 to 275 cm. Two metrics of the PVC curves were used for further quantification: (1) The y-intercept of the curve ($\triangle x = 0$ cm) represents the correlation of population activity at the same corridor position between two days, indicating the cross-session stability of the network. (2) The maximum absolute initial slope of the curve ($0 \le \triangle x \le 100$ cm) represents the reduction of correlation at increasing $\triangle x$, indicating the level of spatial precision of the network. Only sessions 3 days apart were included in this analysis.

To understand if the loss of periodicity in neuronal population activity in Figure A4 simply reflects behavioral changes, we performed a detailed analysis of the periodicity of the population vector corre-lation (PVC) curves shown in Fig. 4A. Results are shown in Figure S4. As a quantifiable metric for the periodicity, we chose the relative peak prominence (RPP), which is the relative height of each peak compared to its neighboring valleys. Example curves and their associated RPPs are displayed in Figure S4A with a quantification in Figure S4B. Fig-ure S4C shows that although there is a measurable positive correlation between RPP and mean licks per trial when considering sessions in the late post-stroke period as well as the entire dataset, the relationship is weak ($R^2 < 0.1$), and suggests that the loss of periodicity does not simply reflect changes in lick rates.

**Bayesian decoder.** A Bayesian decoder was developed to assess the predictability of corridor position based on neural activity. All models used the top 100 neurons with the highest within-session stability that

were present in the training and decoding datasets. To construct the probability function for the decoder, spatial activity maps of training trials for all neurons were smoothed with a Gaussian kernel ($\sigma = 5$ cm) and the mean and standard deviation (SD) across all training trials were computed. Spatial bins with a cross-trial SD below the average per-bin SD of that neuron had their SD set to the mean SD. The estimated position of the animal $\hat{x}$ at time $t$ using the neural activity of $N$ neurons was defined as:

$$\hat{x}(t) = argmax_x \left( occ(x) \prod_{i=1}^{N} \frac{p_i(x,t)}{max_x(p_i(x,t))} \right), where \qquad (4)$$

$$p_i(x,t) = \frac{1}{\sqrt{2\pi\sigma_i(x)^2}} \exp\left( -\frac{(S_i(t) - \mu_i(x))^2}{2\sigma_i(x)^2} \right) \qquad (5)$$

where $occ(x)$ is the occupancy probability per spatial bin $x$ in the training dataset, $\mu_i(x)$ is the mean and $\sigma_i(x)$ is the SD of the spatial activity map at bin $x$ of neuron $i$ in the training dataset, and $S_i(t)$ is the average $\Delta F/F$ value of neuron $i$ in the decoding dataset over time $t \pm \triangle t$, with $\triangle t = 0.5$ s.

For within-session decoding, we adopted a leave-one-out cross-validation approach, where each trial was decoded once while the remaining trials within that session were utilized to create the training set. The final decoder performance for the session was the average metric across all iterations. For cross-session decoding, the training set was generated from the entire last pre-stroke session, and the decoding executed on the entire second session. Two different error metrics were used to quantify the decoder performance: (1) Accuracy is the fraction of correct position bin predictions out of all time points $t$. (2) Sensitivity measures the fraction of time points $t$ out of all time points where the animal was inside a RZ and the decoder correctly predicted to be in a RZ. Chance levels of decoder error metrics were empirically determined by randomly shuffling the position bins in the training dataset 500 times, using the shuffled training set to decode the position of the respective trial, and averaging the error metrics of each session across all animals.

**Analysis of pairwise firing activity between neurons.** Activity synchronicity within neuronal populations was analyzed by computing the pairwise Pearson's correlation coefficients of the fluorescence traces of all neurons within each network. Correlation of the $\Delta F/F$ traces indicates the synchronicity of neuronal activity in time and is termed"pairwise firing activity". Correlation of the spatial activity maps yields the synchronicity of neuronal activity in the linear corridor, and is termed "spatial synchronicity". A cell pair was considered "highly synchronous" if it had a high correlation value for correlated activity or a spatial synchronicity value within the 95th percentile of the network.

Cross-phase changes of pairwise firing activity between neurons[59] were investigated by computing the pairwise Pearson's correlation coefficient of $\Delta F/F$ traces for each session and averaging the coefficients of each cell pair across the sessions within each experimental phase. Neurons that were not present at all sessions during the compared pair of periods were excluded from this analysis. These phase-averaged pairwise correlation coefficients were matched across phase pairs, yielding three distributions (Healthy – Early post, Healthy – Late post, Early post – Late Post) of matched cell pairs for each mouse. For quantification, the Pearson's correlation coefficient, corresponding p-value, and slope of a linear regression model were computed for each distributions using SciPy's linregress function. Distributions with a p-value exceeding 0.05, indicating nonsignificant correlation, were excluded from the results.

## Sensorimotor tasks
To assess possible deficits in the sensorimotor system after microsphere injection, animals were tested in several behavioral tasks before the injection to establish a baseline, and re-tested 2, 7, 10, 22, 32 and 35 days after microstrokes.

## Neurological deficit score
Mice were tested for neurological deficits using a common scoring system[48,60]. In brief, possible deficits in different motor functions (limb clasping, C-shape bending, forepaw grasping and hindlimb repositioning) were scored with 0 (no deficit), 1 (moderate deficit) or 2 (severe deficit). Individual scores of each test were added to yield a total neurological deficit score.

## Skilled forelimb grasping
A grasping test was performed to assess skilled forelimb function using the MotoTrak system[61] (Vulintus). The setup consisted of a lever behind a acrylic glass plate, which was only accessible for the animal with one forelimb through a narrow gap in the plastic. Mice were trained to reach for the lever and pull it towards them to receive a water reward, with sessions lasting 10 – 15 min. An attempt of the animal to reach for the lever was considered a trial. If the pulling force exceeded 5 g, the trial was counted as a "hit"; if the mouse touched the lever, but without enough force, the trial was considered a "miss"; if the forelimb missed the lever, the trial was discarded. The performance per session was quantified as the hit-ratio $\frac{hits}{trials}$ *100.

## Open Field test
To test if microstrokes could impair mobility locomotion, mice were allowed to freely explore an empty open-roofed box of 40×40×40 cm for 6 min, once before and once in the first week after stroke. The movement of the animal was recorded with a video camera and the trajectory extracted with the EthoVision software[62].

## Histology
After conclusion of all experiments (6–8 weeks after microsphere injection), mice were deeply anesthetized with 5% Isoflurane and overdosed with pentobarbital (Kantonsapotheke Zurich, 300 mg/kg body weight, i.p. injection). As soon as respiratory arrest occurred, 0.05 mL Heparin (Braun) was injected into the left ventricle and the animal was perfused transcardially with cold 0.1 M PO4 followed by 4% paraformaldehyde (PFA) in 0.1 M $PO_4$. Brains were extracted, post-fixed (4% PFA, 4 °C, 24 h), cryoprotected (30% sucrose, 0.1 M $PO_4$, 4 °C, 48 h), embedded in OCT (Tissue-Tek, Sakura), frozen at -80 °C, and 100 µm coronal sections were cut with a sliding cryostat (Microm HM 560). Free-floating slices were stored in cryoprotectant (30% ethylenglycol, 15% sucrose, 0.003% Na-azide in 0.1 M $PO_4$). Every third slice (excluding cerebellum) was selected for immunostaining. Sections were washed 3 × 10 min in 1× PBS at room temperature (RT), incubated in blocking solution (10% natural donkey serum (NDS), 0.3% Triton X-100, 1× PBS) for 24 h at 4 °C, followed by an incubation with guinea pig anti-GFAP (1:750, Synaptic Systems, Cat. No. 173004), rabbit anti-Iba1 (1:500, Wako, Cat. No. 019-19471) or rat anti-CD68 (FA-11 clone, 1:500, Invitrogen, Cat. No. 14-0681-82) in antibody buffer (10% NDS, 0.1% Triton X-100, 1× PBS) for 72 h at 4 °C. Slices were washed 3 ×10 min in 1× PBS at RT, then incubated in secondary antibodies (Cy3-conjugated donkey anti-guinea pig, donkey anti-rabbit or donkey anti-rat, 1:250, Jackson) for 4 h at RT. Slices were exposed to DAPI (1:1000), washed again in 1× PBS and mounted on Superfrost Plus glass slides (Thermo Fisher Scientific) with fluorescent mounting medium (Dako). The whole-slice images were acquired with an Axio Scan.Z1 automatic slide scanner (10× objective, 0.6 µm/px resolution, extended depth of field; Zeiss).

## Image analysis

Images were analyzed with QuPath v0.2.3[63] for microsphere and lesion detection. Experimenters were blinded to the experimental condition of each mouse, and microspheres were manually counted and assigned to brain regions using the P56 Allen Mouse Brain Reference Atlas. The non-zero sphere counts of sham-operated mice served as an estimate of the Type I error rate (false positives) of microsphere detection.

For a subset of brains, lesions around microspheres were quantified via fluorescent markers of astrocyte (GFAP) and microglia (Iba1, CD68) activation. If the average fluorescence of GFAP, Iba1, CD68 or autofluorescence (Cy5 channel, signaling cell debris) of the area surrounding a microsphere was at least 2 standard deviations higher than the same area in the contralateral hemisphere, this area was labeled as "lesioned". Similarly, if an area not directly associated with a visible microsphere showed significant fluorescence increase compared to the contralateral hemisphere as well as control slices (slices of the same region from sham-injected animals), the area was labeled as "lesioned". For each brain, microsphere numbers and lesion volumes were summed for each region, and the total microsphere load for the whole brain was estimated by multiplying the counts and volumes by the fraction of imaged volume of each brain based on average volumetric data provided by the Allen Mouse Brain CCFv3[64].

To investigate if microsphere abundance in specific brain regions affected VR task performance, we built two generalized linear models (GLMs) using statsmodels[65]. The dependent variables were average VR task performance during early and late post-stroke phases relative to healthy baseline performance. Independent variables were microsphere counts in a set of brain regions (hippocampus, neocortex, striatum, thalamus, white matter, other), which collectively cover the entire brain. We used a Gamma distribution for microsphere counts and the identity link function, assuming an additive effect of the number of microsphere on task performance. As the Gamma distribution assumes non-negative values, and some neural metrics could be negative, we performed Ordinary Least Squares (OLS) regression to investigate the impact of sphere location on neural metrics.

## Data management and statistical analysis

Experimental data and analysis pipelines were managed by a custom DataJoint database[66] (RRID:SCR\_014543) implemented in Python v3.7. Statistical analysis and plotting was performed with Python and Prism v10 (Graphpad), while figures were assembled with Adobe Illustrator (v28.3). Unless stated otherwise, data are reported as mean ± standard deviation (SD), and data points in figures represent individual animals, with analyses performed per session and averaged across days of each experimental phase. Boxplots are drawn with the box extending from the 25th to 75th percentiles, and the middle line plotted at the median. Whiskers reach to the minimum and maximum values of the distribution. Details regarding the statistical tests employed, multiple hypothesis correction, and the use of repeated-measures statistical testing are outlined in the figure captions and listed in Supplementary Table 1 providing exact p-values.

## Data and materials availability

Raw and processed data are available on the data platform DANDI: https://doi.org/10.48324/dandi.001184/0.240829.1458.

All remaining data are available in the manuscript or the supplementary materials. For individual figures source data are provided with this paper.

## Reporting summary

Further information on research design is available in the Nature Portfolio Reporting Summary linked to this article.

## Code availability

The codes used for the processing and analysis of the raw data are made available as a GitHub repository - https://doi.org/10.5281/zenodo.14906653 - and with more example data on a google drive: https://drive.google.com/drive/folders/1ab3ikKsryVG2jC4eWNAiDETfdJxLhd-J.

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

## Acknowledgements

We thank Hansjörg Kasper for technical advice and fruitful discussions as well as Antonia Weingart for support with illustrations. We thank Anna Schmidt-Rohr for programming the first GUI for tracking neurons in different experimental sessions. We thank Philipp Bethge for providing transgenic mice used in this study. This study was supported by the Branco Weiss Fellowship, the Novartis foundation for biomedical research, Dementia Research/Synapsis Foundation Switzerland, the Hurka Foundation, the TRR 274 – Checkpoints of Central Nervous System Recovery by the DFG and the ERC Starting Grant ARISE awarded to A.S.W as well as a PhD fellowship of the Center for Neuroscience Zurich awarded to H.H.

## Author contributions

A.S.W. ideated and provided the concept. H.H. and A.S.W. designed the study. H.H., A.S.W. and V.I. performed surgeries and carried out experiments. A.S.W., A.R. and M.W. designed and developed the virtual reality corridor. H.H., F.K., V.I., J.G. and A.S.W. performed the data analysis with input from F.H. F.H. provided resources. H.H. and A.S.W prepared figures and wrote the manuscript with input from all authors.

## Funding

## Competing interests

The authors declare no competing interests.
