## [Peer Review file · Nature Communications]

Brain-wide microstrokes affect the stability of memory circuits in the hippocampus

Corresponding Author: Professor Anna-Sophia Wahl

Version 0:

Reviewer comments:

Reviewer #1

(Remarks to the Author)

The manuscript of Heiser et al describes longitudinal 2-photon imaging of calcium activity of CA1 pyramidal cells in the hippocampus of mice subject to brain-wide microstrokes. The experimental and sham control mice were trained to run along a VR corridor to receive water rewards and spatial coding on both the single cell and population level were assessed before and after stroke induction. The authors found a dose-dependent effect of microstrokes on spatial cognition and corresponding changes to the fidelity of spatial coding in the hippocampus. These included a decrease in the stability of both single place fields and population vectors, changes in network structure and a loss of synchronous activity. Overall, the experiment is well designed, and the data are presented clearly. I have several suggestions that I would like the authors to consider:

- 1) The authors coin the terms “functional imprinting” to describe neurons that remain stable place cells, or alternatively, shift to non-coding cells. This is not a standard term in the field, and this general phenomenon is typically considered to reflect population drift. I recommend not coining a new phrase here, as additional jargon is not required to describe this observation.
- 2) The example mouse behavior shown in Fig. 1E reflects a lack of task engagement 4 days after stroke (almost no licking anywhere on the track); is the drop in task performance due to the mouse licking in the wrong location or simply not licking at all? Is the total lick rate similar across recovery in the stroke mice that perform >75%? Please describe the behavioral results in more detail, as this is key to interpreting the changes in network activity. Specifically, is the loss of periodicity in Fig 4A simply reflecting behavioral changes?
- 3) 25 mice across three genotypes were used; some transgenically expressing GCaMP, others receiving a viral injection. Were these genotypes counterbalanced across groups? Please provide a supplementary table listing all mice, genotype, sex, group (Sham or Stroke), outcome (recovery/no recovery) and number of cells imaged per session. The size of the simultaneously imaged population in individual animals is important to know to interpret the PV and decoding data.

(Remarks on code availability)

Reviewer #2

(Remarks to the Author)

This manuscript describes intriguing results on how diffuse cortical infarcts, which may not directly impair motor/sensory function can change ‘cognitive’ function – in this case spatial navigation and prediction of zones of reward. Specifically, they use longitudinal 2-photon imaging to examine the impact of “microstrokes” on hippocampal (CA1) networks as mice navigate a virtual reality environment. The underlying clinical problem that this study addresses is important, as multifocal ischemic strokes are common, and the mechanisms by which such pattern of distributed injury results in cognitive impairment are poorly understood. The authors show interesting evidence that behavioral performance in the VR task appears to be impaired following stroke in a manner that scales with the burden of microsphere delivery (and, in a correlated fashion, overall stroke volume). The aspect that needs clarification is the neural analysis and link to behavior. For example, they classify as two types of recovery – but it seems more of a continuum rather than dichotomous (Fig 3A). The criterion listed for learning/recovery in the methods seem a bit arbitrary. Moreover, as listed below more clarification is needed about the methods. Overall, it is an interesting manuscript, that with revisions, can make an important contribution.

- The bit based quantification is quite opaque. A supplementary devoted to this is key. For example, in fig 1e what are the respective bit rates?
- Fig 1E – it almost appears that ‘early’ they are still relatively accurate and then late – there is way too many licks. Is this consistent in animals? Might suggest a more complex evolving phenotype than simply loss of function. Also suggests the need to interpret the bit rate carefully.
- The performance criteria appear to be somewhat arbitrary (e.g. line 870-883). Why was 75% used as a threshold? How was early and late defined?
- How was a ‘no recovery animal’ defined? The curves in blue and red in Fig 3 A show some overlap. For example, two of the ‘no recovery’ show improvements and then perhaps a decline. How do time-varying changes during recovery map onto the phenomenon identified here.
- Fig 1F/G: What does the x axis mean for healthy?
- Figure 2A: what is meant by a lost cell? Insufficient description in body of paper and figure legend.
- Figure 2: why is a range of n listed for stroke?
- Figure 4: PVC curves appear flatter for the no-recovery group, even in the first column (healthy-healthy); does the lack of recovery reflect, in some part, the nature of the network correlations prior to stroke?
- Figure 5: I am not convinced that the data support claims that the authors are making in this figure. Before looking at top 95th percentile, there were no significant differences in delta F/F correlation. Moreover, in early post-stroke, when there are the most deficits in the recovered group, there are no significant differences in “close” or “in” when compared to sham. Even further, the biggest change in late stroke is emergence of highly correlated PCs in “close” but not “in.” I think the paper might be stronger without this figure (or with a complete revision of this figure).
- Distribution of stroke volume would be helpful to see. Are these distributions different between recovered and non-recovered animals? I.e., do non-recovered animals simply have more strokes of same size, or do they also have larger strokes?
- Can this method of stroke delivery result in retinal ischemia? This would be important to know since performance in the VR task driven predominantly by visual navigation.
- Lesions are induced exclusively in the left hemisphere and imaging is done in ipsilateral CA1. Rationale for this is not discussed.
- The stroke model does not closely resemble any naturally occurring ischemic stroke pattern. Small vessel ischemic disease results in widely distributed microinfarcts and is associated with cognitive dysfunction in humans, though these lesions are predominantly subcortical and bilateral. Some discussion regarding these caveats is warranted in a limitation sections at the end.

(Remarks on code availability)

Reviewer #3

(Remarks to the Author)
Comments to Authors

In this manuscript, Heiser et al. studied the effect of brain-wide microstrokes on hippocampal place cell representations in mice. The authors used chronic two-photon calcium imaging to observe the response of hippocampal CA1 pyramidal cells during virtual reality navigation before and after induced strokes in mice by injection of microspheres. In their task, mice learned an operant virtual reality navigation task that requires licking for water reward in four reward zones on a linear track. For both before stroke and after stroke, the authors studied the stability of functional coding of pyramidal cells in hippocampal CA1. They also separated mice with stroke induction into recovery and no-recovery groups and investigated the network properties and synchronized firing of CA1 neurons in these two conditions. Overall, the authors presented clear evidence on how microstrokes 1) reduce the number of stable place cells and increase the number of non-coding cells, and 2) disrupt transition probability between functional classes. For animals that behaviorally recover from the stroke, they show higher network stability and increased synchronized firing between cell pairs as compared to the no-recovery group.

This is a well-executed study with careful and comprehensive analyses. The manuscript presents clear experiment data and analyses, and presents convincing evidence regarding cellular and network-level disruptions following strokes. The work investigates an important topic and begins to fill a major knowledge gap. There are a few points that require the author’s careful attention to enhance further the impact of their impressive work:

1. One main suggestion to the authors is that in their analyses, they should separately look at place cell stability near and far from the reward zone. At this stage, it is not very clear to what extent cross-day stability is driven by cell near reward. Previous studies (e.g., Grosmark et al., Nature Neuroscience, 2021) found that place cells near reward have higher cross-day stability. Relatedly, a minor point that the authors may consider acknowledging chronic two-photon imaging study (Grosmark et al., above) and also chronic imaging under diseased conditions (Zaremba et al., Nature Neuroscience, 2017) in their Discussion about chronic activity monitoring in the hippocampus (Lines 409-413). Further relatedly, in Figure 5, the definition of “far”, and “close” regions isn’t convincing. It would be better to define those regions by the relative distance to reward locations rather than the current way. That is, the “close” regions should be centered around reward zones, while “far” regions should then surround the “close” regions.

2. Behavior paradigm. In the manuscript, the authors described the behavior and imaging paradigm using daily imaging for 5 days before stroke induction, and starting from 3 days post-stroke, mice were imaged every third day. It is unclear whether the animals were trained after stroke induction and in between imaging days. In my opinion, not training the mice between imaging days can cause behavior fluctuations. As evident in panels 1J,K&L, the proportions of stable, unstable, and non-coding cells show different trends of changes even in sham mice.

3. Functional imprinting. The authors first compared the day-to-day transition probability between functional cell classes, and then calculated the same metric for two groups of animal post-stroke induction. While the before-stroke data were true day to day as the animals are imaged every day, the post point would be comparing imaging sessions that were performed at least two days apart. Similar to my previous point, it is unclear if animals were trained in between recording days in post-stroke, and thus this metric can have more variation. The authors could consider comparing every third day for recordings before stroke induction.

4. Functional Connectivity. The authors described their calculation of the correlation between pairs of neurons as functional connectivity. This was mentioned in Figure 5 legends, discussion and methods section but not mentioned in the main text. In my opinion it is ok to say the inferred functional connectivity in this case as a way to measure the co-firing or proximity firing between two neurons. However, for both clarity and consistency, the authors should describe this metric simply as the pairwise firing between neurons instead of functional connectivity.

5. Location and volume of the lesion in relation to hippocampal pyramidal cell coding properties. In figure S1, the authors performed a clear analysis of post-hoc histological sections to identify the location of microsphere deposits and the volume of lesion. They also described in the main text that most microspheres and lesions were located in the neocortex, hippocampus, and subcortical regions such as the thalamus and striatum. They further quantified the effect on task performance based on lesion location in S1F. In my opinion, an analysis that focuses on the correlation of lesion location and volume to the same metrics analyzed in Figure 2,3,4&5 is needed. The authors could identify whether lesions in certain brain regions are more likely to cause changes in hippocampal circuits. Similarly, it is unclear why certain mice recover behaviorally following lesions. It could also be the difference in the location and volume of where the microspheres were deposited.

Minor

1. The reference needs a thorough reorganization with full attention to scholarahip. Several papers are listed twice (e.g., citations 12 and 13: Hainmuller and Bartos, Nature, 2018).
2. Sham mice should receive a saline injection. As currently states in the methods section, it is unclear whether the Sham mice only underwent surgery or received the vehicle injection.
3. In Figure 2, instead of plotting true data and shuffled data side-by-side, authors could consider plotting the difference between true and shuffled data as a summary. The figure in its current form is very confusing to understand.
4. In Figure 4 Panel E & F, label x and y axes.
5. In lines 247, 248 and 264, please specify Figure 3C, 3D, 3E and 3F.
6. In line 283, please specify which figure.
7. In line 295, please specify which figure.
8. In line 429, there should be a comma after 'task'.

(Remarks on code availability)

Version 1:

Reviewer comments:

Reviewer #1

(Remarks to the Author)

The authors have done a great job revising the manuscript and have addressed all my concerns. I only have 2 very minor suggestions, both on page 6 of the manuscript:

In 198- i believe the word "remaining" should be "retaining"

In 218- I think the word "On" should be "In"

(Remarks on code availability)

Reviewer #2

(Remarks to the Author)

The authors have responded in a thorough manner. No concerns remain.

(Remarks on code availability)

Reviewer #3

(Remarks to the Author)

The manuscript is further improved after a thorough revision by the authors. All my comments were satisfactorily addressed. I have no further points.

(Remarks on code availability)

Our point-to point response to the concerns of the individual reviewers:

We would first like to thank all reviewers for their thorough study of our manuscript as evidenced by their constructive criticism, and their generally positive feedback. Below, we address each reviewer's individual comments.

REVIEWER COMMENTS

Reviewer #1 (Remarks to the Author):

The manuscript of Heiser et al describes longitudinal 2-photon imaging of calcium activity of CA1 pyramidal cells in the hippocampus of mice subject to brain-wide microstrokes. The experimental and sham control mice were trained to run along a VR corridor to receive water rewards and spatial coding on both the single cell and population level were assessed before and after stroke induction. The authors found a dose-dependent effect of microstrokes on spatial cognition and corresponding changes to the fidelity of spatial coding in the hippocampus. These included a decrease in the stability of both single place fields and population vectors, changes in network structure and a loss of synchronous activity. Overall, the experiment is well designed, and the data are presented clearly. I have several suggestions that I would like the authors to consider:

- 1) The authors coin the terms “functional imprinting” to describe neurons that remain stable place cells, or alternatively, shift to non-coding cells. This is not a standard term in the field, and this general phenomenon is typically considered to reflect population drift. I recommend not coining a new phrase here, as additional jargon is not required to describe this observation.**

We thank the reviewer for his comment. We have removed the term “functional imprinting” throughout the revised manuscript, in particular on page 5/6 and page 11. We instead used terms such as “place cell turn-over” (Shuman et al., 2020) or “functional cellular identity” or “functional determination of neurons”, which are self-explanatory.

- 2) The example mouse behavior shown in Fig. 1E reflects a lack of task engagement 4 days after stroke (almost no licking anywhere on the track); is the drop in task performance due to the mouse licking in the wrong location or simply not licking at all? Is the total lick rate similar across recovery in the stroke mice that perform >75%?**

In the revised form of the manuscript we further examined if the drop in task performance shown in Fig.1E reflects “licking in the wrong location” or “not licking at all”. We thus plotted in Supplementary Figure S2, E the “mean number of licks per trial in the VR corridor across experimental groups and stages. Microstrokes did not affect licking behavior, as sham and stroke mice did not show significant differences before or after surgery”. Furthermore, we show in Supplementary Figure S2, F that there are some changes in the licking rate within individual mice (the Sham and Recovery groups show a reduced lick rate from Healthy to Early Post-stroke, and the Recovery group increases the lick rate between Early and Late post-stroke periods), but a relationship with the task performance is not apparent. Importantly, animals in the No Recovery group, which show the most pronounced task performance deficits, do not change their lick rate. We therefore argue that the observed performance changes are not purely due to a lack of task engagement. We thus included in the results section the following sentence (page 4 of the revised manuscript): “Notably, microstrokes did not affect the licking

rate itself of experimental groups, as sham and stroke mice did not show significant differences before and after stroke induction (Supplementary Figure 2E, F).”

Please describe the behavioral results in more detail, as this is key to interpreting the changes in network activity. Specifically, is the loss of periodicity in Fig 4A simply reflecting behavioral changes?

Besides the new quantification of the lick rates during the navigation in the VR corridor (Supplementary Figure 2E, F), we also included exemplary data (Figure S7B) of an animal that displayed extensive anticipatory licking during the expert phase to show the heterogenous nature how animals mastered the spatial navigation task to identify reward zones. Our behavioral analysis using a spatial information (SI)-based performance metric reflects the focused licking pattern also with this strategy, while a simpler metric of the percentage of licks within reward zones (lick hits) could not accurately represent the performance of the animal. We have included this explanation, why we used a SI based analysis to quantify the behavior across different strategies in the methods section (page 27) stating: “As the strategies how to identify reward zones in the spatial navigation task varied among animals and six animals (IDs 33, 69, 91, 93, 110 and 122, see table S2) showed extensive anticipatory licking in front of reward zones (Figure S7B) also during expert phases, we used the spatial information (SI) content of the lick histogram. The SI-based performance metric also better reflected the focused licking patterns of such strategies, while simpler metrics such as the percentage of licks within reward zones (lick hits) could not accurately represent the performance of the animal. To quantify the spatial information (SI) content, we first created a lick histogram by binning licks (defined as the time points when the tongue of the animal touched the water spout) into 120 spatial bins, and computing the lick probability (the fraction of trials with at least one lick) for each position bin (**Error! Reference source not found.E**).”

To understand if the loss of periodicity in neuronal population activity in Figure A4 simply reflects behavioral changes, we performed a detailed analysis of the periodicity of the population vector correlation (PVC) curves shown in Figure 4A and depict the results in a new supplementary figure (Figure S4). As a quantifiable metric for the periodicity, we chose the relative peak prominence (RPP), which is the relative height of each peak compared to its neighboring valleys. Example curves and their associated RPPs are displayed in Figure S4A with a quantification in Figure S4B. Figure S4C shows that although there is a measurable positive correlation between RPP and mean licks per trial when considering sessions in the late post-stroke period as well as the entire dataset, the relationship is weak ($R^2 < 0.1$), and suggests that the loss of periodicity does not simply reflect changes in lick rates. We included this statement on page 31 of the methods section.

We also included in the results section (page 8): “The loss of periodicity in the No Recovery mice did not simply reflect changes in behavior measured in lick rates as shown in Figure S2E, F and Figure S4.”

3) 25 mice across three genotypes were used; some transgenically expressing GCaMP, others receiving a viral injection. Were these genotypes counterbalanced across groups? Please provide a supplementary table listing all mice, genotype, sex, group (Sham or Stroke), outcome (recovery/no recovery) and number of cells imaged per session. The size of the simultaneously imaged population in individual animals is important to know to interpret the PV and decoding data.

We added the requested data in the Supplementary Table S2. The “Stroke” group is split into the outcome groups “Recovery” and “No Recovery”, and a dedicated “Group” column has been omitted. The number of imaged neurons per session is presented for each animal as a

list in square brackets. We have included the information for this supplementary table on page 25 of the revised manuscript.

Reviewer #2 (Remarks to the Author):

This manuscript describes intriguing results on how diffuse cortical infarcts, which may not directly impair motor/sensory function can change ‘cognitive’ function – in this case spatial navigation and prediction of zones of reward. Specifically, they use longitudinal 2-photon imaging to examine the impact of “microstrokes” on hippocampal (CA1) networks as mice navigate a virtual reality environment. The underlying clinical problem that this study addresses is important, as multifocal ischemic strokes are common, and the mechanisms by which such pattern of distributed injury results in cognitive impairment are poorly understood. The authors show interesting evidence that behavioral performance in the VR task appears to be impaired following stroke in a manner that scales with the burden of microsphere delivery (and, in a correlated fashion, overall stroke volume). The aspect that needs clarification is the neural analysis and link to behavior. For example, they classify as two types of recovery – but it seems more of a continuum rather than dichotomous (Fig 3A). The criterion listed for learning/recovery in the methods seem a bit arbitrary. Moreover, as listed below more clarification is needed about the methods. Overall, it is an interesting manuscript, that with revisions, can make an important contribution.

- 1) The bit based quantification is quite opaque. A supplementary devoted to this is key. For example, in fig 1e what are the respective bit rates?**

We thank this reviewer for the thoughtful comments. To clarify the interpretation of the task performance quantification metric, we have now added the spatial information (SI) content in bits to each histogram in Figure 1E: The better the performance, the higher the performance value in bits.

We chose this bit-based quantification over a simpler metric of counting how many licks were inside reward zones (lick hits) because it better reflected the performance of animals with strategies to identify reward zones and thus different licking behavior. A majority of mice had lick patterns as those shown in the exemplary histograms in figure 1E. However, since our task did not actively punish licks outside of reward zones, six animals (IDs 33, 69, 91, 93, 110 and 122, see table S2) showed anticipatory licking in front of reward zones, where lick rates would quickly ramp up right before entering a reward zone. If we would only calculate a simple ratio between licks in and outside of reward zone as read-out for the cognitive performance, animals with anticipatory licking would receive most deficient scores although the anticipatory licking pattern suggests that the animal can anticipate the approaching reward, and thus reflects an understanding of the spatial structure of the task. The bit-based metric however values the spatial context of each lick, which, in our opinion, more accurately represents the performance of the animal within the spatial navigation task with and without anticipatory licking.

To illustrate the advantage of the SI content over the simple “lick hit” metric in quantifying the performance of animals with anticipatory licking, we added panel B to the new supplementary figure S7B, which is analogous to figure 1E, but for a mouse with anticipatory licking. While the histogram of the “Expert” session shows an understanding of the task, the “lick hit” metric is lower than during the “Naïve” and “3 day” session, while the SI content reflects the increased spatial precision of the licks during the “Expert” session more accurately.

We have also included the explanation, why we used a SI based analysis to quantify the behavior across different strategies in the methods section (page 27) stating: “As the strategies how to identify reward zones in the spatial navigation task varied among animals and six animals (IDs 33, 69, 91, 93, 110 and 122, see table S2) showed extensive anticipatory licking in front of reward zones (Figure S7B) also during expert phases, we used the spatial information (SI) content of the lick histogram. The SI-based performance metric also better reflected the focused licking patterns of such strategies, while simpler metrics such as the percentage of licks within reward zones (lick hits) could not accurately represent the performance of the animal. To quantify the spatial information (SI) content, we first created a lick histogram by binning licks (defined as the time points when the tongue of the animal touched the water spout) into 120 spatial bins, and computing the lick probability (the fraction of trials with at least one lick) for each position bin (**Error! Reference source not found.E**).”

2) Fig 1E – it almost appears that ‘early’ they are still relatively accurate and then late – there is way too many licks. Is this consistent in animals? Might suggest a more complex evolving phenotype than simply loss of function. Also suggests the need to interpret the bit rate carefully.

In the revised form of the manuscript we further examined if the drop in task performance shown in Fig.1E reflects “licking in the wrong location” or “not licking at all”. We thus plotted in Supplementary Figure S2, E the “mean number of licks per trial in the VR corridor across experimental groups and stages. Microstrokes did not affect licking behavior, as sham and stroke mice did not show significant differences before or after surgery.” Furthermore, we show in Supplementary Figure S2, F that there are some changes in the licking rate within individual mice (the Sham and Recovery groups show a reduced lick rate from Healthy to Early Post-stroke, and the Recovery group increases the lick rate between Early and Late Post-stroke periods), but a relationship with the task performance is not apparent. Importantly, animals in the No Recovery group, which show the most pronounced task performance deficits, do not change their lick rate. We thus included in the results section the following sentence (page 4 of the revised manuscript): “Notably, microstrokes did not affect the licking rate itself of experimental groups, as sham and stroke mice did not show significant differences before and after stroke induction (Supplementary Figure 2E, F).”

3) The performance criteria appear to be somewhat arbitrary (e.g. line 870-883). Why was 75% used as a threshold? How was early and late defined?

The early and late post-stroke phases were defined following the translation of the “acute” and “subacute” phase to rodent models of stroke, which are suggested to encompass the first week and the first month after stroke e.g. according to Bernhardt et al., 2017 and Joy & Carmichael, 2021. This timeline appeared to match our observations, as animals that did not recover spontaneously within the first week after stroke would not show any significant improvement of task performance for the rest of the experiment (see Figure 3A).

Small fluctuations in performance are normal across animals. To quantify this, we computed the standard deviation (SD) of task performance between all mice for each expert-level session, with the average SD across all sessions being 23.8%. When rounding to 25%, we get a threshold of 75%, where performance deficits below 75% can be considered to be outside the normal performance fluctuation range. To visualize the distribution of task performance changes after stroke, we added the new supplementary Figure S7A, which

shows a scatter plot of the mean task performance (compared to baseline) of all animals during early and late post-stroke.

We added these details to the method section of the manuscript (page 28).

4) How was a ‘no recovery animal’ defined? The curves in blue and red in Fig 3 A show some overlap. For example, two of the ‘no recovery’ show improvements and then perhaps a decline. How do time-varying changes during recovery map onto the phenomenon identified here.

Following the description in the manuscript (lines 915-917), a mouse was defined as “No Recovery” if its average task performance in both early post-stroke and late post-stroke sessions was below 75% of the healthy baseline (see also Figure S7A). As mentioned in the response to the previous question, some fluctuation of task performance across multiple days is normal for these kinds of complex behavioral paradigm. While some animals in the “No Recovery” group indeed show some improved performance on individual days, these improvements are not consistent across the post-stroke period and significantly lower compared to the Recovery group.

Furthermore, we investigated the animal that we assume this reviewer referred to more closely (the “No Recovery” animal with a comparably high performance on days 12 and 15 in figure 3A). This animal (number 110, see Table S2) had the highest fraction of stable place cells (figure 3B), and the best decoder performance (figure 3D-F) during the late post-stroke period in the “No Recovery” group. Furthermore, this was the only animal in the “No Recovery” group with a low number of microspheres in the brain (Figure S1C). We thus discussed these findings in the discussion section on page 10: “Supporting this idea, we found also in the No Recovery group animals with some degree of cognitive improvement (although not in the range of the Recovery group). These animals had the highest fraction of stable place cells (Figure 3B), the best decoder performance (Figure 3D-F) in the late post-stroke period, but also the lowest number of microspheres in the brain (Figure S1C) relative to the other animals of this group, suggesting that our experimental model provided a continuum of stroke severity and ability for recovery, correlated to the total load of microstrokes in the brain.”

5) Fig 1F/G: What does the x axis mean for healthy?

In the scatter plots, the x-axis refers to the number of microspheres found in the brains during post-mortem histological analysis. The “healthy” dataset represents data from animals before injection of microspheres and the data points should serve as a control to rule out potential biases in the healthy condition, such as animals with a low baseline task performance receiving more microspheres by chance. We revised the figure legend Fig 1F accordingly, now stating: “Healthy” represent data points from mice before microsphere injection serving as controls”.

6) Figure 2A: what is meant by a lost cell? Insufficient description in body of paper and figure legend.

A „lost” cell is a neuron which was initially imaged in the healthy condition, but could not be tracked any more due to major tissue modifications and edema after stroke. We initially included these cells in the Sankey diagram for visual reasons to have the bars add up to 100% of all tracked cells, but these lost cells were not included in any analysis, as we do not have

any further information about the functionality of these cells during these sessions. Thus, we removed these lost cells now from Figure 2A to avoid further confusion for the reader.

7) Figure 2: why is a range of n listed for stroke?

Some animals, especially in the “No Recovery” group, did not have any place cells recorded for some post-stroke sessions, which means that place cell transition probabilities could not be computed. These animals are omitted from the boxplot of the respective experimental stage.

8) Figure 4: PVC curves appear flatter for the no-recovery group, even in the first column (healthy-healthy); does the lack of recovery reflect, in some part, the nature of the network correlations prior to stroke?

We thank for this hint and performed an additional quantification of the “flatness” of the PVC curves and defined it as the relative height difference between the peaks and their surrounding valleys (relative peak prominence, RPP), reported in Figure S4. We show that while the curves of the “No Recovery” group during the healthy period may appear flatter by eye, there is no significant difference between healthy sessions across the experimental groups (Figure S4B).

9) Figure 5: I am not convinced that the data support claims that the authors are making in this figure. Before looking at top 95th percentile, there were no significant differences in delta F/F correlation. Moreover, in early post-stroke, when there are the most deficits in the recovered group, there are no significant differences in “close” or “in” when compared to sham. Even further, the biggest change in late stroke is emergence of highly correlated PCs in “close” but not “in.” I think the paper might be stronger without this figure (or with a complete revision of this figure).

We believe that the key findings of this figure were not clearly presented in the first version of the manuscript. Thus, we fully revised this figure, adding new data to the figure and within the supplement. The data presented in Figure 5 and the supplement (new Supplementary Figures S5 and S6) provide answers to 1) how pairs of surviving neurons in rewiring networks behave after stroke, 2) if there are functional classes of cells active at distinct locations which are more or less important for the cognitive outcome after stroke and 3) if there is a spatial relationship (cells are physically located next to each other) between cells, which are co-active in the rewiring network after structure, suggesting structural rewiring processes of neurons nearby.

The average pairwise $\Delta F/F$ correlation of the whole network calculating the mean correlation between all pairs of neurons per network did not reveal a significant difference between experimental groups after stroke (Figure 5B). However, we identified a different composition of functional cell classes, when comparing activity of neuronal pairs with high and low functional connectivity. When plotting the cumulative distribution of the functional connectivity of place cell and non-coding cell pairs, we observed a clear difference for the 95th percentile of neuronal pairs (with highest functional connectivity) between place cells and non-coding cells in the healthy condition (Figure 5C). Highly correlated cells (the 95th percentile at the dashed red line) included more place cells than non-coding cells, justifying a subsequent quantification of the composition of functional cell types in this 95% percentile subset of highly correlated cell pairs. We then examined how the composition of this 95% percentile changed after stroke. We found in animals of the No-Recovery group, a reduced percentage of synchronous active place cells early after insult (Figure 5D), while animals with a recovery from the cognitive deficits (Recovery group) revealed an increase of the percentage of synchronous active place cells after stroke (Figure 5D). Consecutively, the percentage of non-coding pairs of neurons with highly correlated activity was pronounced in animals with a

chronic cognitive deficit compared to sham animals after stroke (Figure S5B). We next asked if neurons with highly correlated activity also share the same spatial representation. Corresponding to the results in Figure 5D, we also found a decline of spatial correlation in animals of the No-Recovery group early after stroke (Figure 5E). We then examined whether these highly correlated functional cells were clustered around salient locations such as reward zones. A clustering of place fields around reward zones has been reported repeatedly (Markus et al., 1995, J Neurosci; Hollup et al., 2001, J Neurosci; Dupret et al., 2010, Nat Neurosci; Grosmark et al., 2021, Nat Neurosci). However, in our experiment with chronic imaging before and after microstroke induction we found that place fields of neurons with highly correlated activity were uniformly distributed in the corridor, except for animals of the Recovery group during the later post-stroke state (Figure 5F, G). The increased number of place fields close, but not in reward zones (Figure 5G), may be related to the phenomenon of anticipatory licking (see supplementary figure S7B) and the reward prediction mechanism of the dopaminergic system, where the reward response is triggered by an associated stimulus (such as the visual cue of the approaching distinctive wall patterns of the reward zone).

We also examined to which extent cross-day stability of place cells is driven by cells active near reward zones. We first analysed the average distance to the next reward zone of stable and unstable place cells, split by experimental group and stage. Plotting these data as a distance difference, with positive values indicating that unstable place cells are closer to reward zones, and negative values that stable place cells are closer to reward zones suggested no connection between place cell stability and distance to reward, as all datasets were not significantly different from 0 (Figure S6A). We also correlated place field distance to the next reward zone to the cross-session stability of place cells across experimental groups and stages (Figure S6B). While we found no significant correlation for animals with a stroke, sham mice displayed weak but significant negative correlation between cross-session stability and place field distance during early and late post-stroke sessions.

We also analysed if neurons with highly correlated activity are physically located close to each other in the rewiring network after stroke (Figure S5A). We found no specific relationship between synchronous active cells and their (Euclidean) distance to each other in the chronically imaged networks.

We included all these findings in the fully revised text and Figure 5/Figures S5 and S6 in the revised version of the manuscript. We also discussed our findings in the discussion section on page 13 stating: “A clustering of place fields around reward zones has been reported repeatedly^{24,38–40}. However, in our experiment with chronic imaging before and after microstroke induction we found that place fields of neurons with highly correlated activity were uniformly distributed in the corridor, except for animals of the Recovery group during the later post-stroke state (Figure 5F, G). The increased number of place fields close, but not in reward zones (Figure 5G), may be related to the phenomenon of anticipatory licking (Figure S7B) and the reward prediction mechanism of the dopaminergic system, where the reward response is triggered by an associated stimulus (such as the visual cue of the approaching distinctive wall patterns of the reward zone).”

We also examined if the stability of place cells is driven by neuronal activity near reward zones as previous studies²⁴ revealed that place cells near reward zones have higher cross-day stability. In our data we found only weak evidence for a higher stability of place cells close to reward zones. A possible explanation may be that in our experiments mice were well accustomed to the corridor, as they had been exposed to the same context for up to 2 months prior. In contrast, most studies use novel or changing corridors, and context exposure is often limited to a few days or weeks. It is possible that the relationship between place field reward

proximity and cross-session stability is more pronounced during the early learning phase of the corridor, but the bias is slowly replaced during learning by a more uniform distribution, a phenomenon already suggested on shorter time scales by Grosmark et al.,²⁴. Future experiments with our setup may include imaging sessions during the learning period, which might provide further information about the temporal dynamics of these mechanisms.

10) Distribution of stroke volume would be helpful to see. Are these distributions different between recovered and non-recovered animals? I.e., do non-recovered animals simply have more strokes of same size, or do they also have larger strokes?

We thank the reviewer for this intriguing question which we had not considered before. We thus performed another analysis (Supplementary Figure S1D) showing that the No Recovery group did not only have more strokes than the Recovery group, but also slightly (on average + 3 μm^3) larger strokes. The cause of this finding is not immediately obvious, as both groups received injections of spheres with the same diameter. The effect size is small, but the larger stroke volume might have contributed to impaired task performance and recovery, in addition to the higher stroke load. We thus included in the results section: “Notably, No-Recovery animals had also significantly larger lesions than Recovery mice in post-mortem histological analysis (Figure S1D).”

11) Can this method of stroke delivery result in retinal ischemia? This would be important to know since performance in the VR task driven predominantly by visual navigation.

Despite the injection of microspheres in the internal carotid artery, several collateral anastomoses with the open A. carotis externa exist (via the A. meningea media, A. temporalis superficialis and A. angularis), enabling the arterial perfusion of the retina also in our model. However, as we did not collect retinas post-mortem from our experimental animals we cannot fully exclude retinal damage. Nevertheless, we could investigate whether the task performance would correlate with the percentage of microspheres that were located within visual cortical areas (acronym “VIS” in the Allen P56 Mouse Reference Atlas). As shown in the new panel G in supplementary figure S1, there is no correlation between task performance and microspheres in visual cortex during early poststroke, and a negative, but non-significant correlation during late poststroke. While this result cannot exclude the possibility of retinal ischemia, it does suggest that there is no direct link between damage to the visual system and task performance, especially during the early post stroke phase when the most severe performance deficits occur. We thus stated on page 4 of the results section: “As our spatial navigation task involved in particular visual cues in the VR-setup to identify reward zones, we also analyzed the relative amount of microspheres in visual areas (following the Allen Brain Atlas as reference). We found no significant correlation between the number of microspheres stuck in visual areas and the task performance after stroke, suggesting no direct link between damage to the visual system and task performance (Figure S1G).”

12) Lesions are induced exclusively in the left hemisphere and imaging is done in ipsilateral CA1. Rationale for this is not discussed.

We have several reasons for only injecting microspheres to the left hemisphere and performing imaging in the corresponding left hippocampus: (1) Bilateral injection of microspheres in both internal carotid arteries lead to the death of >90% of animals. (2) Simultaneous imaging of ROIs in both hippocampi using chronic 2photon imaging in the hippocampus is currently technically impossible. (3) We were in particular interested in

understanding the effect of microspheres in the mainly affected circuits which in particular involved the ipsilesional hippocampus. However, future studies should reveal processes in the healthy, contralesional hippocampus after microstroke induction, e.g. examining structural changes of newly rewired bilateral hippocampal circuitry using tracing technology.

We discussed these issues under “Limitations” on page 14 of the revised manuscript.

13) The stroke model does not closely resemble any naturally occurring ischemic stroke pattern. Small vessel ischemic disease results in widely distributed microinfarcts and is associated with cognitive dysfunction in humans, though these lesions are predominantly subcortical and bilateral. Some discussion regarding these caveats is warranted in a limitation sections at the end.

We thank the reviewer for this hint. We have now included a “Limitations” section on page 14 stating: “With the here presented microstroke model we aimed at mimicking key features of small vessel disease, which comprises the majority of cases of vascular dementia⁴¹. This includes the induction of disseminated brain-wide microstrokes, a measurable cognitive deficit and an effect on hippocampal networks. However, while in the human presentation of small vessel disease microstrokes are predominantly located subcortically, we found in our stroke model widely distributed microstrokes, in particular also in cortical regions. As bilateral microsphere injections in the common carotid arteries in mice lead in most cases to fatal outcomes⁴², we chose unilateral injections of microspheres in the left common artery leading to microstrokes predominantly in the left hemisphere, where we imaged the effect of the disseminated microstrokes on left hippocampal networks. However, future studies should reveal the effects of the microstrokes on the contralateral hippocampus using multi-area imaging and tracing technology to reveal also structural changes of newly rewired bilateral hippocampal circuitry in relation to the cognitive outcome.”

Reviewer #3 (Remarks to the Author):

Comments to Authors

In this manuscript, Heiser et al. studied the effect of brain-wide microstrokes on hippocampal place cell representations in mice. The authors used chronic two-photon calcium imaging to observe the response of hippocampal CA1 pyramidal cells during virtual reality navigation before and after induced strokes in mice by injection of microspheres. In their task, mice learned an operant virtual reality navigation task that requires licking for water reward in four reward zones on a linear track. For both before stroke and after stroke, the authors studied the stability of functional coding of pyramidal cells in hippocampal CA1. They also separated mice with stroke induction into recovery and no-recovery groups and investigated the network properties and synchronized firing of CA1 neurons in these two conditions. Overall, the authors presented clear evidence on how microstrokes 1) reduce the number of stable place cells and increase the number of non-coding cells, and 2) disrupt transition probability between functional classes. For animals that behaviorally recover from the stroke, they show higher network stability and increased synchronized firing between cell pairs as compared to the no-recovery group.

This is a well-executed study with careful and comprehensive analyses. The manuscript presents clear experiment data and analyses, and presents convincing evidence regarding cellular and network-level disruptions following strokes. The work investigates an important topic and begins to fill a major knowledge gap. There are a

few points that require the author's careful attention to enhance further the impact of their impressive work:

- 1) One main suggestion to the authors is that in their analyses, they should separately look at place cell stability near and far from the reward zone. At this stage, it is not very clear to what extent cross-day stability is driven by cell near reward. Previous studies (e.g., Grosmark et al., Nature Neuroscience, 2021) found that place cells near reward have higher cross-day stability. Relatedly, a minor point that the authors may consider acknowledging chronic two-photon imaging study (Grosmark et al., above) and also chronic imaging under diseased conditions (Zaremba et al., Nature Neuroscience, 2017) in their Discussion about chronic activity monitoring in the hippocampus (Lines 409-413). Further relatedly, in Figure 5, the definition of "far", and "close" regions isn't convincing. It would be better to define those regions by the relative distance to reward locations rather than the current way. That is, the "close" regions should be centered around reward zones, while "far" regions should then surround the "close" regions.**

We would like to thank this reviewer for his thoughtful comments. We have now included the recommended literature concerning chronic imaging in health and disease in the revised manuscript.

To address the questions concerning the distance to reward zones on the stability of place cells, we performed two additional analyses shown in a new supplementary figure S6. First, we analysed the average distance to the next reward zone of stable and unstable place cells, split by experimental group and stage. We plot this data as a distance difference, with positive values indicating that unstable place cells are closer to reward zones, and negative values that stable place cells are closer to reward zones (Figure S6A). The results suggest that there is no connection between place cell stability and distance to reward zones, as all datasets are not significantly different from 0. Second, we took a more unbiased approach and correlated place field distance to the next reward zone to the cross-session stability of place cells across experimental groups and stages (Figure S6B). Most datasets show no significant correlation, and only the Sham group displays a weak but significant negative correlation between cross-session stability and place field distance during early and late post-stroke sessions.

Thus, our results provide only weak evidence for a higher stability of place cells close to reward zones, which has been reported more clearly by the studies cited by the reviewer. A possible explanation may be that during the analysed imaging sessions, the animals are well accustomed to the corridor, as they have been exposed to the same context for up to 2 months prior. In contrast, most studies use novel or changing corridors, and context exposure is often limited to a few days or weeks. It is possible that the relationship between place field reward proximity and cross-session stability is more pronounced during the early learning phase of the corridor, but the bias is slowly replaced during learning by a more uniform distribution, a phenomenon already suggested on shorter time scales by Grosmark et al., 2021. Future experiments with our setup may include imaging sessions during the learning period, which might provide further information about the temporal dynamics of these mechanisms. We have included these issues on page 13 ff in the "Discussion" section of the manuscript.

In Figure 5 we defined the zones as "far" and "close" in relation to the distance to the next reward zone instead of the relative distance to the closest reward zone because we think that the future reward zone is more relevant for the animals than the passed reward zone in our experimental set-up. The screens of our VR setup are located in front of and to the sides of the mouse, meaning that the passed sections of the corridor are not displayed, but the next

reward zone is already visible from the current reward zone. The observed lick patterns (Figure 1E, Figure S7B) support this assumption, as expert mice lick right before and in the beginning of the reward zone, and very rarely at the end or after the zone. Finally, we found that the location of the place fields e.g. for Recovery animals (Figure 5F) is clustered right before and not after reward zones. These results suggest that for the animals the absolute distance to the next reward zone is more relevant than the relative distance to the closest (passed or upcoming) reward zone.

- 2) Behavior paradigm. In the manuscript, the authors described the behavior and imaging paradigm using daily imaging for 5 days before stroke induction, and starting from 3 days post-stroke, mice were imaged every third day. It is unclear whether the animals were trained after stroke induction and in between imaging days. In my opinion, not training the mice between imaging days can cause behavior fluctuations. As evident in panels 1J,K&L, the proportions of stable, unstable, and non-coding cells show different trends of changes even in sham mice.**

We did not train animals in between imaging days, meaning that the animals only got exposed to the corridor every 3rd day after stroke induction. The primary rationale for this training protocol was to avoid potentially neuro-rehabilitative over-training, as we wanted to observe the pure effect of microstrokes on hippocampal networks and mechanisms of spontaneous recovery. A possible increase of behavior fluctuations should affect all mice equally, and may have at most added noise to the data, which would increase the relevance of the observed statistical significances (Figure 3A). We thus argue that the potential danger of over-training outweighs the possibility of increased behavior fluctuations. We thank the reviewer for the hint that this information was missing in the methods section. We now added the relevant description in the “Methods” section of the revised manuscript (page 29) stating: “Microstrokes were induced after the last baseline imaging session on the same day (day 0), followed by post-stroke imaging sessions every third day up to 4 weeks after stroke to study the “pure” effect of the disseminated lesions on the hippocampal network and to avoid neurocognitive training/rehabilitation induced plasticity. Animals were not exposed to the corridor outside of imaging sessions.”

- 3) Functional imprinting. The authors first compared the day-to-day transition probability between functional cell classes, and then calculated the same metric for two groups of animal post-stroke induction. While the before-stroke data were true day to day as the animals are imaged every day, the post point would be comparing imaging sessions that were performed at least two days apart. Similar to my previous point, it is unclear if animals were trained in between recording days in post-stroke, and thus this metric can have more variation. The authors could consider comparing every third day for recordings before stroke induction.**

Figure 2B primarily aims at characterizing healthy baseline dynamics using daily recordings. Instead, figure 2C focuses on early and late post-stroke phases independently, where imaging sessions were consistently performed every third day. Although the increased temporal distance in post-stroke compared to healthy sessions might slightly decrease the observed effect size, however this does not abolish the imprinting phenomenon. For example, place cell-place cells- transitions exhibit similar transition probabilities in the healthy ($13.49 \pm 10.85\%$) and early post-stroke phases ($12.64 \pm 8.67\%$). Importantly, the phenomenon remains clearly measurable in sham mice, even with the three-day interval (Figure 2C). Since the pre-

stroke data and post-stroke analyses are not directly compared, we argue that the temporal resolution differences do not impact our primary conclusions of this data.

Furthermore, when directly comparing cross-session metrics across phases (e.g. for the PVC analysis in Figure 4), we did only include sessions that were 3 days apart, also before the stroke induction, as the reviewer suggests. However, this detail was not stated in the manuscript, and we thank the reviewer for reporting these omissions in this and the previous question. The relevant information has been added in the revised manuscript on page 29.

- 4) Functional Connectivity. The authors described their calculation of the correlation between pairs of neurons as functional connectivity. This was mentioned in Figure 5 legends, discussion and methods section but not mentioned in the main text. In my opinion it is ok to say the inferred functional connectivity in this case as a way to measure the co-firing or proximity firing between two neurons. However, for both clarity and consistency, the authors should describe this metric simply as the pairwise firing between neurons instead of functional connectivity.**

We thank the reviewer for this hint. To avoid any confusion for the reader we replaced the term “functional connectivity” in the text and replaced it - as suggested - by “pairwise firing between neurons” or “correlated activity between pairs of neurons” in the main text, methods and figure legends.

- 5) Location and volume of the lesion in relation to hippocampal pyramidal cell coding properties. In figure S1, the authors performed a clear analysis of post-hoc histological sections to identify the location of microsphere deposits and the volume of lesion. They also described in the main text that most microspheres and lesions were located in the neocortex, hippocampus, and subcortical regions such as the thalamus and striatum. They further quantified the effect on task performance based on lesion location in S1F. In my opinion, an analysis that focuses on the correlation of lesion location and volume to the same metrics analyzed in Figure 2,3,4&5 is needed. The authors could identify whether lesions in certain brain regions are more likely to cause changes in hippocampal circuits. Similarly, it is unclear why certain mice recover behaviorally following lesions. It could also be the difference in the location and volume of where the microspheres were deposited.**

We performed the requested analysis by examining the effect of microsphere location on several different markers of neural circuit activity (percentage of stable place cells, spatial discrimination (via population vector correlation), and position decoder accuracy) with Ordinary Least Squares regression, shown in supplementary figure S1I. Similar to the results for task performance, no region showed a clear association with neural metrics. White matter had the largest coefficients, but this result may be biased due to the small volume of white matter tracts and thus few microspheres and a small sample size, as suggested by the large error bars.

It is indeed not completely clear why some mice recover behaviorally, while others do not. However, as mentioned in the first paragraph of the “discussion” section, our data suggests that the total number of microspheres as well as lesion volume impacts the ability of mice to recover (figure S1C and D). Furthermore, we did not see an effect of sphere location on task performance (figure S1F) during early nor late post-stroke, which we would expect if stroke location was affecting behavioral recovery.

Minor

- 1) The reference needs a thorough reorganization with full attention to scholarship. Several papers are listed twice (e.g., citations 12 and 13: Hainmuller and Bartos, Nature, 2018).**

We thank the reviewer for noticing this mistake. The reference section has been carefully revised.

- 2) Sham mice should receive a saline injection. As currently states in the methods section, it is unclear whether the Sham mice only underwent surgery or received the vehicle injection.**

Injections in the internal carotid artery are difficult procedures where the common carotid artery and the external carotid artery are temporarily ligated to prevent major bleedings with fatal outcomes during the injection. Also, after the injection the common carotid artery is compressed for 6-10 min to prevent bleedings. Thus, besides the microsphere injection, the injection procedure itself with the ligation of the common carotid artery can induce disturbed cerebral blood flow. We thus decided against sham animals with saline injections. However, as stated in the methods section we included mice with failed microsphere injections to the sham group as stated on page 29 of the methods section: "Mice that maintained >75% relative VR performance in both post-stroke phases after microsphere injections and did not have significantly more detected spheres than sham-operated mice in their brains were pooled with sham-operated mice for subsequent analysis."

- 3) In Figure 2, instead of plotting true data and shuffled data side-by-side, authors could consider plotting the difference between true and shuffled data as a summary. The figure in its current form is very confusing to understand.**

We thank the reviewer for this reasonable suggestion, as we also realized the unintuitive presentation of the data. The revised figure 2 and supplementary figure S3 should now be much easier to understand and more effective in visualizing the main results of the figure.

- 4) In Figure 4 Panel E & F, label x and y axes.**

The axes labels have been added in the revised figure.

- 5) In lines 247, 248 and 264, please specify Figure 3C, 3D, 3E and 3F.**

The manuscript has been updated accordingly.

- 6) In line 283, please specify which figure.**

The manuscript has been updated accordingly.

- 7) In line 295, please specify which figure.**

The manuscript has been updated accordingly.

- 8) In line 429, there should be a comma after 'task'.**

The manuscript has been updated accordingly.

Our point-to point response to the concerns of the individual reviewers:

We would like to thank all reviewers for their thorough study of our manuscript as evidenced by their constructive criticism, and their positive feedback. Below, we address the final comments of reviewer Reviewer #1.

Reviewer #1 (Remarks to the Author):

**The authors have done a great job revising the manuscript and have addressed all my concerns. I only have 2 very minor suggestions, both on page 6 of the manuscript:
In 198- i believe the word "remaining" should be "retaining"
In 218- I think the word "On" should be "In"**

We thank for this hint and have revised the two phrases in line 198 and line 217 of the revised manuscript.

Reviewer #2 (Remarks to the Author):

The authors have responded in a thorough manner. No concerns remain.

Reviewer #3 (Remarks to the Author):

The manuscript is further improved after a thorough revision by the authors. All my comments were satisfactorily addressed. I have no further points.